# Regulatory basis for reproductive flexibility in a meningitis-causing fungal pathogen

Pengjie Hu[1,6], Hao Ding[1,2,6], Huimin Liu[1,3,6], Yulin Yang[1,2], Lei Chen[1,2], Guang-Jun He[1], Weixin Ke[1,2], Ping Zhu[4], Xiuyun Tian[1], Yan Peng[4], Zhenghao Shen[1,2], Xiaoxia Yao[1,2], Changyu Tao ⓟ[5], Ence Yang ⓟ[5], Guojian Liao[4], Xiao Liu ⓟ[1,2] & Linqi Wang ⓟ[1,2,3] ✉

Pathogenic fungi of the genus *Cryptococcus* can undergo two sexual cycles, involving either bisexual diploidization (after fusion of haploid cells of different mating type) or unisexual diploidization (by autodiploidization of a single cell). Here, we construct a gene-deletion library for 111 transcription factor genes in *Cryptococcus deneoformans*, and explore the roles of these regulatory networks in the two reproductive modes. We show that transcription factors crucial for bisexual syngamy induce the expression of known mating determinants as well as other conserved genes of unknown function. Deletion of one of these genes, which we term *FMP1*, leads to defects in bisexual reproduction in *C. deneoformans*, its sister species *Cryptococcus neoformans*, and the ascomycete *Neurospora crassa*. Furthermore, we show that a recently evolved regulatory cascade mediates pre-meiotic unisexual autodiploidization, supporting that this reproductive process is a recent evolutionary innovation. Our findings indicate that genetic circuits with different evolutionary ages govern hallmark events distinguishing unisexual and bisexual reproduction in *Cryptococcus*.

Unlike pathogenic bacteria, many eukaryotic microbial pathogens, such as fungi or protozoan parasites, are capable of using sexual reproduction to facilitate their infections and adaptations[1–16]. The benefits of sexual reproduction occur through different mechanisms. First, sexual meiotic recombination generates genetic diversity in progeny, enabling lineage advantages related to emergence of hyper-virulent and drug-resistant variants[1–5,9,13–16]. Second, sexual or parasexual cycles can result in karyotypic plasticity and offspring aneuploidy, which are important mechanisms leading to antimicrobial resistance in clinical and environmental isolates[2,5,13,17–19]. Third, the sexual differentiation program often produces specific morphologies that are advantageous for pathogen colonization and survival in the host[6–8,10].

These benefits created by sexual reproduction have all been described in *Cryptococcus* pathogens, which represent the most important human pathogenic fungi in phylum Basidiomycota[3,5,7,8,13]. These pathogens infect the lungs via haploid airborne sexual spores or yeast and frequently cause fatal meningoencephalitis[6–10,20,21]. Among them, *Cryptococcus deneoformans* (formerly *C. neoformans* var. *neoformans*, serotype D) serves as a model organism for the study of *Cryptococcus* sexual reproduction, due to the robustness of its sexual reproduction which occurs with clearly identifiable stages[22–24]. *C. deneoformans* has two mating types (α and **a**) and can undergo different sexual cycles: α-**a** bisexual reproduction and unisexual reproduction (also named haploid fruiting), which involves only a single mating type (mainly α)[23,25–28]. Owing to an extreme bias of α in natural populations (>99%) that likely results in infrequent bisexual mating, α

[1]State Key Laboratory of Mycology, Institute of Microbiology, Chinese Academy of Sciences, 100101 Beijing, China. [2]University of Chinese Academy of Sciences, 100049 Beijing, China. [3]University of Science and Technology of China (USTC), 230026 Hefei, China. [4]The Medical Research Institute, College of Pharmaceutical Sciences, Southwest University, 400715 Chongqing, China. [5]Department of Microbiology, School of Basic Medical Sciences, Peking University Health Science Center, 100191 Beijing, China. [6]These authors contributed equally: Pengjie Hu, Hao Ding, Huimin Liu. ✉e-mail: wanglq@im.ac.cn

unisexual reproduction is considered to be the dominant sexual form, with implications for evolution of *C. deneoformans* and other important *Cryptococcus* pathogens[3,28–32].

The bisexual and unisexual cycles share multiple developmental and molecular processes, and they are initiated in response to nearly the same environmental cues[27,29]. Upon sexual induction, differentiation of both bisexual and unisexual communities begins with the yeast-to-hypha transition, which produces root (invasive) and aerial hyphae[29,33,34]. This morphological transition takes place in a stochastic manner, resulting in a highly structured mating colony with yeasts predominantly localized in the colony center and hyphae mainly in the periphery[29,33,34]. Tips of a subpopulation of aerial hyphae subsequently differentiate to form a sexual structure (basidium) in which meiosis occurs[21,29,33,34]. Successful spatiotemporal coordination of basidial maturation and meiotic progression leads to the formation of four chains of basidiospores from the basidial apex[21,33,34]. In spite of these similarities during differentiation, these two sexual cycles can be discriminated by the approaches to achieve pre-meiotic diploidization (PD): bisexual PD involves formation of α/**a** zygotes via α-**a** cell-cell fusion (syngamy), while unisexual PD is mainly accomplished by autodiploidization, which likely occurs via endoreplication[27,29,35,36].

Due to the importance of sexual reproduction in the life cycle, evolution and pathogenicity of *Cryptococcus* pathogens, great efforts have been undertaken over the last three decades to identify related genes and mechanisms[1,2,6,9,11,26–30,37–43]. However, the regulatory basis that shapes the different sexual life cycles and enables reproductive flexibility remains largely unknown.

In this study, we constructed a library of strains with deletions of genes coding for transcription factors (TFs) in a reference *C. deneoformans* strain that undergoes robust bisexual and unisexual reproduction. TF mutant strains were assessed quantitatively or semi-quantitatively for eight phenotypic traits, which cover the previously identified morphogenesis events and PD phases throughout two sexual life cycles[21,35,44–46]. Functional analysis of *C. deneoformans* TFs unveiled a large number of novel genotype-phenotype connections related to sexual regulation, offering a comprehensive and unprecedented understanding of the functional TF networks that shape the cryptococcal ability to undergo distinct sexual cycles. We also systematically identified the determinants that govern the bisex-specific and unisex-specific PD phases, and our phylogenetic analysis of these determinates provides key insights into the evolution of the hallmark processes of the two reproductive modes.

## Results

### Construction and functional profiling of a gene-deletion library for transcription factors in *Cryptococcus deneoformans*

TFs are generally the terminal components of various signaling cascades and are critical for a wide range of biological processes[38,47,48]. To systematically identify TFs engaged in the *C. deneoformans* bisexual and unisexual life cycles, we de novo constructed a gene-deletion library for TFs in reference strain XL280α, which undergoes either of the reproductive cycles robustly[22,23]. This advantage allows quantitative or semi-quantitative evaluation of phenotypic traits related to sequential phases throughout the two sexual life cycles (Fig. 1a–c). The putative TFs in XL280α were predicted through analysis of the DNA Binding Domain (DBD) TF prediction database (http://www.supfam. org/SUPERFAMILY/) and annotations of DBDs at the Pfam database (https://pfam.xfam.org/). Though these combined analyses, we retrieved a total of 228 putative TF genes in *C. deneoformans* (Supplementary Data 1). Of these candidates, 115 TFs were selected for further analysis because of their remarkably dynamic expression during unisexual or bisexual development, based on two independent and previously published sets of time-series transcriptomic data[21,49]. Of

these 115 TFs, we successfully constructed gene-knockout mutants for 111 TFs by using large-scale homologous recombination; however, after repeated attempts, we failed to obtain the corresponding mutants for four TFs (Hsf1, Cdc39, Pzf1, and Esa1), whose homologs have been demonstrated to be essential genes in other fungal species[50–54]. To accurately verify the genotype-phenotype connection and exclude any unlinked mutational effects, we generated at least two independent TF mutants for the all 111 TFs, including 4 TFs (Mat2, Znf2, Cqs2, and Cva1) that were constructed previously[46,55], and thus obtained a total of 252 verified mutant strains.

We assessed these TF mutants for eight phenotypic traits related to two mating colony morphogenesis processes, four cellular differentiation stages and two PD stages, as indicated in Fig. 1a. Due to the fact that cellular differentiation stages and colony morphogenesis processes are shared in the two reproduction modes, the corresponding phenotypic traits of the TF mutants were measured only during unisexual development (Fig. 1b, c). For the assessment of most of the community morphogenesis processes and stages of sexual cellular differentiation, we used previously reported methods[21,35,44–46]. In addition, a new quantitative approach was developed in this study for evaluating sporulation (see the Methods section for details).

Our functional profiling revealed a total of 374 genotype-phenotype connections, including 312 associations that had not been previously identified (Figs. 1d and 2a; Supplementary Data 2). The data indicated a marked functional coverage of the TF mutant library, with approximately 91.0% of the TF mutants (101/111) exhibiting at least one detectable phenotype. In addition, we recaptured nearly all of the documented phenotypes of TFs that impact sexual differentiation (mostly filamentation) in *C. deneoformans* or other *Cryptococcus* pathogens, such as *C. neoformans* and *C. gattii*[21,38,44,46,55–64]. These results confirm the reliability of our functional profiling approach and reflect the conserved functions of TFs in sexual reproduction in different *Cryptococcus* pathogens.

Based on the phenome data, we performed a comprehensive network analysis to explore the functional TF networks of various sexual stages (Fig. 2a). Our data revealed two types of TFs: stage-specific TFs that exclusively regulate a single cellular differentiation or PD stage, and pleiotropic TFs that exert their functions in at least two stages (Fig. 2a). The vast majority of identified TFs are pleiotropic regulators, which contribute importantly to the architecture of the TF network of sexual development.

These identified pleiotropic TFs likely play key roles in supporting developmental continuity by coordinating various sexual stages. This idea is supported by the result of a Spearman's rank correlation analysis, which measured the degree of linear dependence between the two possible phenotypic combinations tested in this study. As illustrated in Fig. 2b, this analysis revealed a high inter-correlation between the phenotypes of different sexual processes. In particular, sporulation, the final phase for both reproductive modes, was significantly correlated to all seven other processes (Fig. 2b). This inter-relationship highlights the significance of pleiotropic TFs in integrating different morphogenesis and PD events to ensure the formation of spore (basidiospore) progeny.

In addition, we identified TFs that function differently in the development of invasive or aerial structures, indicating a complexity in spatial regulation of mating colony differentiation (Fig. 2c). We also noticed that all of the TFs controlling colony spatial structure also play roles in cellular differentiation and reproduction mode-specific PD stages (Supplementary Fig. 1). This indicates an importance of these TFs in connecting spatial control of the mating community to the sexual life cycles. Overall, our functional profiling provides a comprehensive view of spatiotemporal regulation of TFs during the multistage sexual cycles that occur in highly differentiated mating communities.

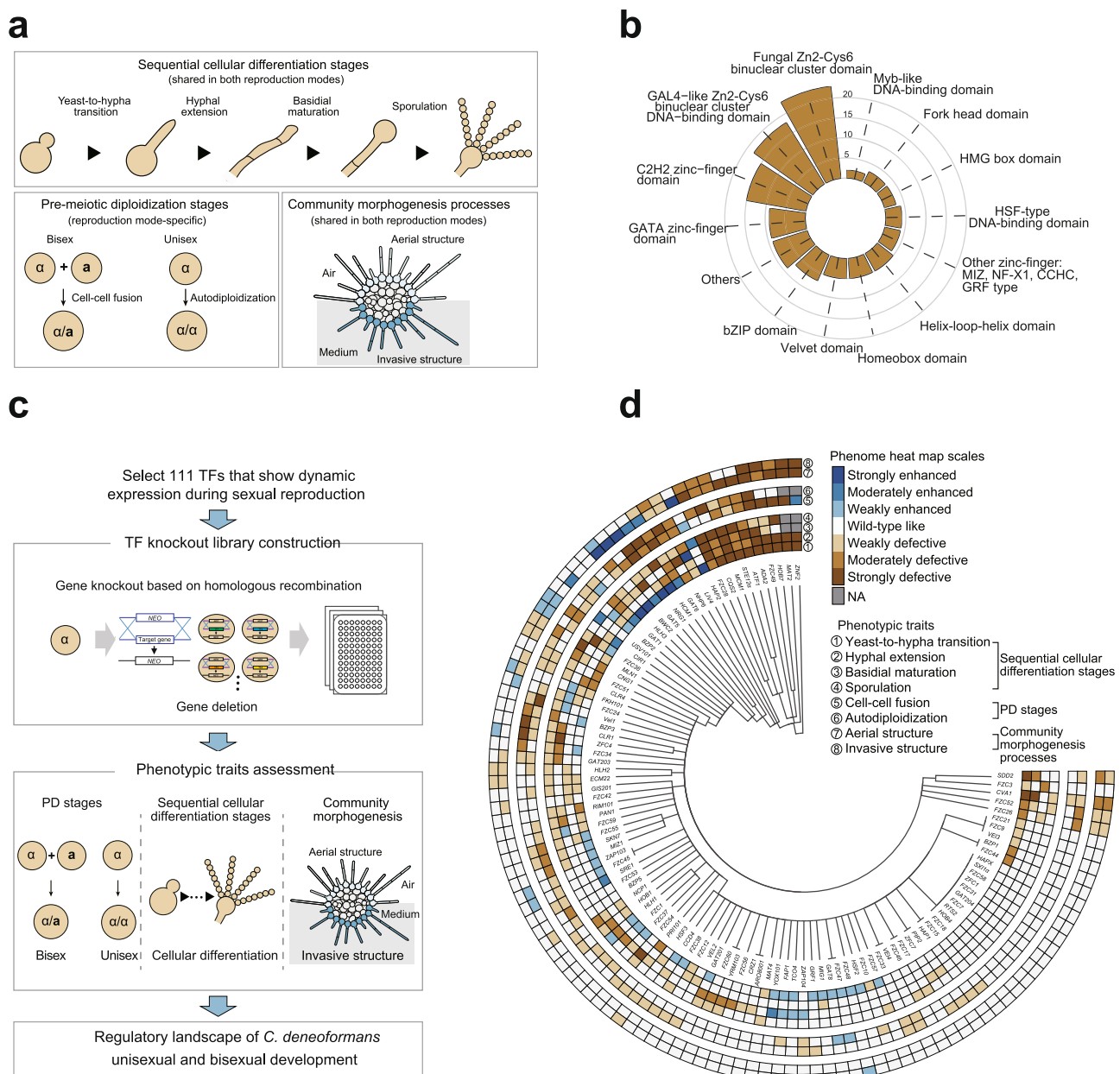

**Fig. 1 | Construction and functional profiling of a gene-deletion library for transcription factors in *C. deneoformans*. a** Schematic diagram showing mating colony morphogenesis and various sexual stages during *C. deneoformans* unisexual or bisexual reproduction. **b** Pie chart showing the classes and distribution of 111 *C. deneoformans* TFs. Each TF was classified based on the DNA binding domains predicted using Superfamily (http://www.supfam.org/SUPERFAMILY/) and Pfam (http://pfam.xfam.org/) databases. DNA binding domain(s) of each TF are listed in Supplementary Date 1. **c** Flowchart of the construction of the *C. deneoformans* TF mutant library and assessment of phenotypic traits during *C. deneoformans* sexual development. **d** Heat map of eight phenotypic traits related to two sexual cycles in *C. deneoformans* TF mutants. Phenotype scores are represented in distinct colors based on quantitative or semi-quantitative phenotypic analyses of ≥2 independent TF mutant strains. Phenotype strengths (weak, intermediate and strong) are distinguished in gradients of red or blue. Quantitative or semi-quantitative phenotypic analyses of phenotypic traits were performed as described in Methods. NA not available. Source data are provided as a Source Data file.

## Identification of TFs with important contributions to the transcriptional response to α-a mating

The functional profiling identified TFs involved in the defining characteristics of two reproduction modes: bisexual syngamy and unisexual autodiploidization. To assess bisexual syngamy, we measured the frequency of α-a cell–cell fusion via unilateral mating between XL280α-derived TF mutants and XL280**a**, a congenic strain of XL280α[22]. In this experiment, 51 TFs displayed detectable impacts on α-a cell–cell fusion (Fig. 1d). These TFs include known mating regulators, such as Mat2, the master regulator of *Cryptococcus* cell-cell fusion[27,29,56]. In addition, we uncovered 32 TFs

(32/51), whose connections to bisexual cell-cell fusion had not been previously reported.

We further selected seven TFs found to be most critical for cell–cell fusion and explored their regulons during mating (Fig. 3a). Of these TFs, only the Mat2 regulon under mating-inducing conditions has been reported[56]. To identify the targets of the other six TFs, we performed transcriptomic analyses via high-coverage RNA sequencing (RNA-seq) and targeted unilateral mating between XL280α-derived TF mutants and the XL280**a** strain (Supplementary Data 3). It was shown that the regulons of seven TFs accounted for approximately 27.5% of the total number of the α-**a** mating-

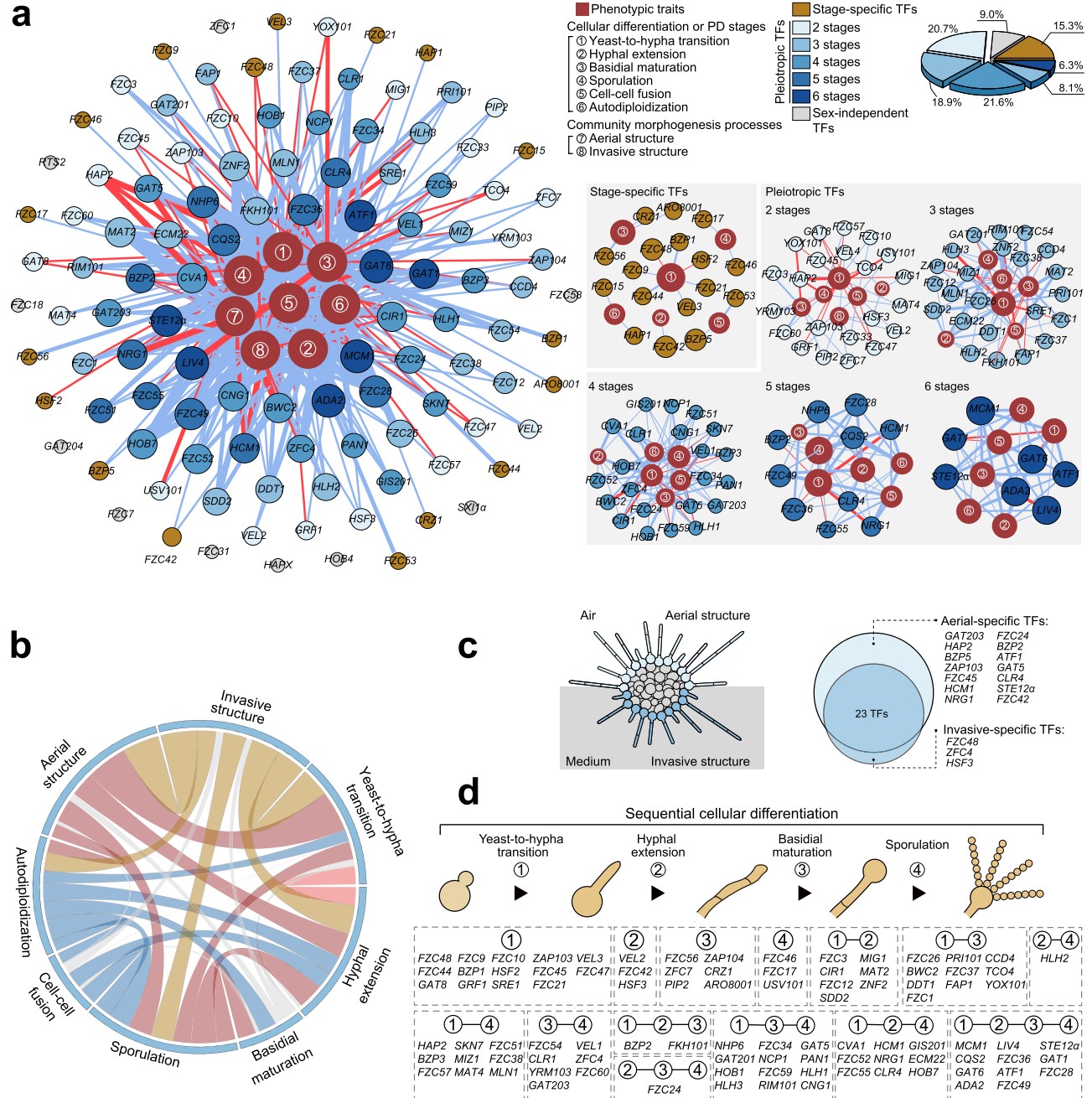

**Fig. 2 | Spatiotemporal functional TF networks underlying sequential stages during sexual community development. a** The bipartite network shows associations between TFs and eight phenotypic traits. Red nodes indicate the eight phenotypic traits, and other nodes represent TFs. Nodes are sized according to their respective degree of connectivity, with bigger nodes indicating higher connectivity. The edges represent the regulation strength (strong, intermediate and weak) of TFs on phenotypic traits, and red and blue edges indicate positive and negative regulations, respectively. A gradient color (from light blue to dark blue) is used to indicate the number of phenotypic traits (1 to 6 stages) regulated by a TF.

A pie chart indicates the percentage of different types of TFs that were classified based on the regulated numbers of stages/events (upper right panel). **b** Consolidated network visualization of correlations among the eight phenotypic traits. The thickness of each link is proportional to the value of the Spearman's *Rho*. Gray edges indicate non-significant associations with *P* > 0.05 (Spearman's rank correlation analysis). **c** A Venn diagram showing TFs that exert spatial control in the development of aerial and invasive structures of sexual colony. **d** TFs are categorized according to their functions in various stage(s) during sequential cellular differentiation. Source data are provided as a Source Data file.

responsive genes (431/1569); these 1569 responsive genes were identified as those with significantly differential expression after the co-culturing of α and **a** cells on mating-inducing medium (Fig. 3b, c). The high degree of representation of the TF regulons among these responsive genes indicates the important involvement of these TFs in the transcriptional response to α-**a** mating. However, a gene accumulation curve indicated the remarkable differences in the mating-responsive genes regulated by these TFs, suggesting that

they may act in concert in mediating transcriptional responses to bisexual syngamy (Fig. 3c). A Gene Ontology (GO) analysis revealed that these TFs regulate a range of processes associated with metabolic or physiological adaptations (Supplementary Fig. 2a). Taken together, these results illustrate the contributions of these TFs to the transcriptional changes that occur in response to mating stimuli; these changes likely result in physiological and metabolic shifts that support bisexual mating.

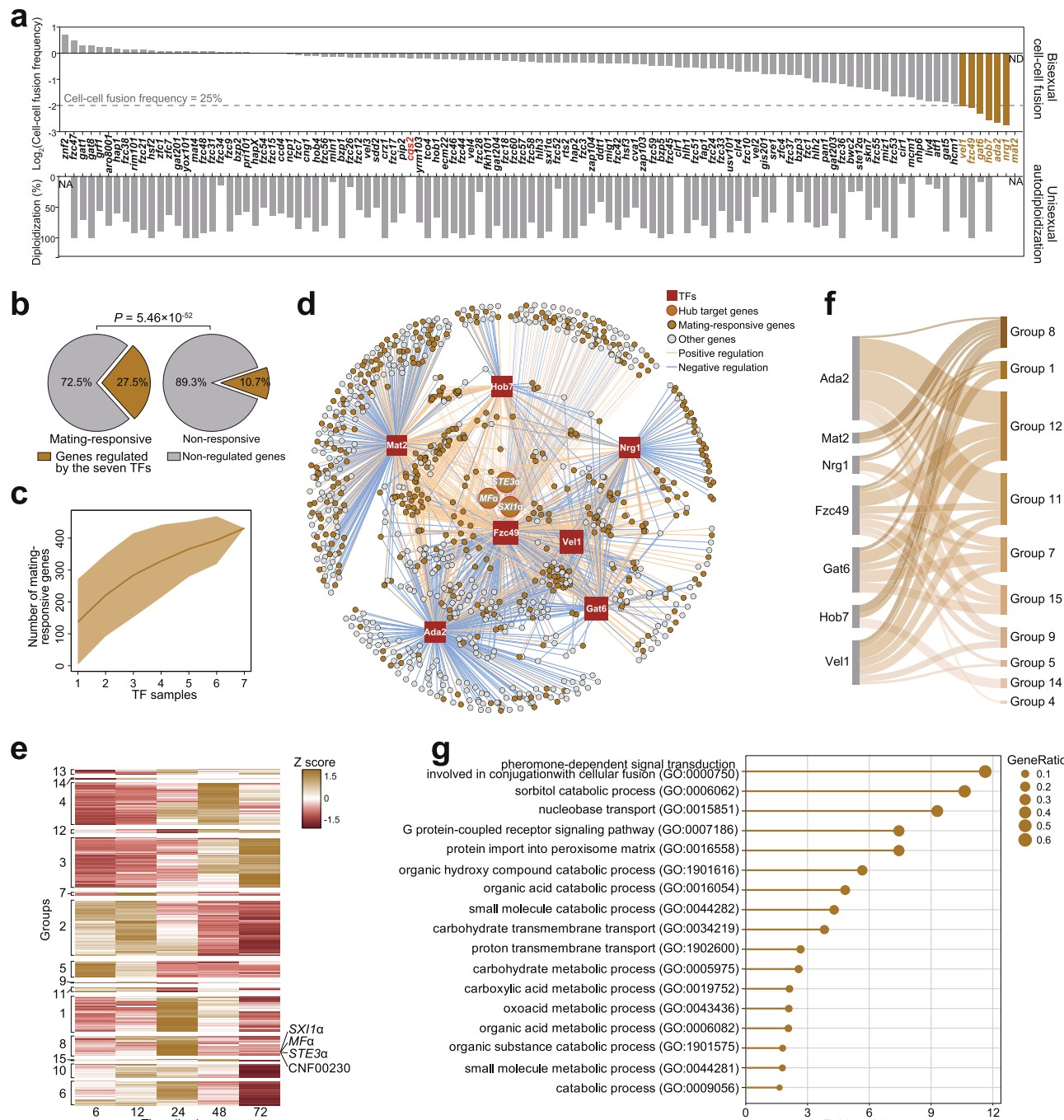

**Fig. 3 | Seven TFs crucial for bisexual syngamy coordinately induce the expression of a specific set of genes centered on ancient mating genes.** **a** Unilateral relative cell-cell fusion frequency and percentage of diploid blastospores of 111 TF deletion mutants. The dashed line indicates 25% cell–cell fusion frequency compared to wild-type strain. The top seven TFs critical for cell-cell fusion are denoted in yellow. To assess autodiploidization, at least six blastospores from different unisexual hyphae of each strain were dissected for FACS-based ploidy detection. Cqs2 (in red) is dispensable for bisexual cell-cell fusion but is strictly required for unisexual autodiploidization. Note that the absence of Mat2 in XL280α completely blocked bisexual cell-cell fusion, and no corresponding fusant was detected. ND, not detected. NA, not available. **b** Percentage of the total targets of the seven TFs among mating-responsive and non-responsive genes. The P value was determined with one-sided Fisher's exact test. **c** Gene accumulation curve for the mating-responsive genes regulated by the seven TFs. The gray line depicts the arithmetic average curve. The shaded area represents the 95% confidence interval.

**d** The regulatory network of syngamy TFs. The RNA-seq data for Ada2, Nrg1, Fzc49, Gat6, Hob7, and Vel1 were generated in this study, while RNA-seq data for Mat2 was obtained from previous work[56]. The square and the circle indicate TFs and their targets, respectively. Yellow and gray circles represent mating-responsive and non-mating responsive genes, respectively. Positive and negative regulation are indicated by yellow and blue edges, respectively. **e** Transcriptional profiles of 15 gene groups identified based on publicly available time-series RNA-seq data targeting different developmental stages after mating induction[21]. Red indicates low expression; yellow indicates high expression. **f** Sankey diagram showing the regulatory connections between the syngamy TFs and the gene groups. Regulatory connections shown were statistically significant (one-sided Fisher's exact test, $P < 0.05$). The Fisher's exact test result is described in more detail in Supplementary Fig. 2d. **g** Gene ontology (GO) analysis of genes in group 8. Source data are provided as a Source Data file.

## Fmp1 serves as a previously unidentified component of the *Cryptococcus* mating MAPK pathway, and its homologs are critically involved in opposite-sex mating in evolutionarily remote fungi

We noted that genes encoding proteins known as sex-determination factors were the only genes significantly upregulated by all seven TFs[26,27,65,66] (Fig. 3d and Supplementary Fig. 2b). These factors include the pheromone Mfα, the pheromone receptor Ste3α and the homeobox regulator Sxi1α, whose homologs are also the conserved components of the mating mitogen-activated protein kinase (MAPK) pathway, an ancient fungal mating signaling cascade[26,27,65,66]. Early studies have proved the hub and specific roles of these sex-determination factors in opposite-sex mating in *Cryptococcus* species or other fungal species[65,67–71]. These roles were verified by our unilateral mating experiments, which showed a complete inability for α-**a** cell-cell fusion in an XL280-derived mutant in which the genes encoding Mfα, Ste3α and Sxi1α were deleted (Supplementary Fig. 2c). By reanalyzing previously published time-series transcriptomic data[21], the coding-genes of the sex-determination factors were shown to be clustered together based on their highly similar expression dynamics during sexual differentiation, again supporting their tight functional connectivity (Fig. 3e). We further identified the genes clustered with the sex-determination genes according to their temporal expression pattern on a genome-wide scale (Fig. 3e). It was found that the resulting gene group (Group 8) was significantly enriched in regulons of all seven mating TFs (Fig. 3f and Supplementary Fig. 2d). Moreover, GO analysis of this group showed an overrepresentation of the genes engaged in pheromone-dependent signal transduction involved in conjugation with cellular fusion (GO: 0000750), indicating the potential importance of this gene group in bisexual cell-cell fusion (Fig. 3g).

We observed that this gene group includes 47 uncharacterized genes, which are highly induced during bisexual mating and whose homologs exist in evolutionarily distant fungal species (Supplementary Data 4). The ages of these genes shown in Supplementary Data 4 were determined based on a previously published phylostratigraphic approach[72]. We hypothesize that these 'ancient' genes may play conserved roles in mating in divergent fungal species. Of them, CNF00230 represents one of the most highly induced genes during α-**a** mating, and its orthologues are widely distributed across species from two main fungal phyla, Ascomycota and Basidiomycota (Fig. 4a and Supplementary Data 5). To test the impact of CNF00230 and its orthologues on fungal opposite-sex mating, these genes were deleted in the basidiomycetes *C. deneoformans* and *C. neoformans* and we also obtained a corresponding gene-deletion mutant from the deletion collection of the ascomycete *Neurospora crassa*[73]. We found that gene deletion led to severe defects of heterosexual reproduction in two *Cryptococcus* pathogens and *N. crassa*, as evidenced by an almost complete abolishing of cell-cell fusion (Fig. 4b, c) and by the formation of perithecia (Fig. 4d). Given the conserved importance of the homologs of CNF00230 in fungal mating, we named it fungal mating-evoking protein 1 (*FMP1*).

Functional domain prediction based on the Pfam program revealed that PH, RING and vWA domains are harbored in the protein encoded by *FMP1* (Fig. 4e). Notably, all of these domains are included in a similar arrangement in Ste5, which is an important component of the mating MAPK pathway in *Saccharomyces cerevisiae*[74–76]. However, reciprocal BLAST searches did not find their orthologous relationship, suggesting that Fmp1 and Ste5 are evolutionarily remote proteins or are not homologous to each other. Supporting this relationship, the structure of the vWA domain, which is essential for the function of Ste5[75], is markedly different from that of the corresponding domain in Fmp1 as predicted by the AlphaFold 2 program (Supplementary Fig. 3). Furthermore, the species that harbor the homologs of *FMP1* are more divergent than the species that harbor *STE5* (Fig. 4a and Supplementary Data 5), which is found only in *Saccharomycotina*[77].

In *S. cerevisiae*, Ste5 is required for signaling through the mating MAPK cascade during syngamy[74–76,78]. Despite disparity in their protein sequences and structures, Fmp1 may have a functional similarity to Ste5, because the mutant lacking Fmp1 exhibited phenotypes similar to those observed in mutants lacking the constituents of the mating MAPK cascade, including Ste11α, Ste7, and Cpk1[9] (Fig. 4b, f). These phenotypes involve defective filamentation and a dramatically attenuated cell-cell fusion efficiency during opposite-sex mating (Fig. 4b, f). Moreover, transcriptomic assays indicated that more than 91% of genes affected by Fmp1 during mating were also regulated by Cpk1, the terminal kinase of the *Cryptococcus* mating MAPK cascade[9] (Fig. 4g, h and Supplementary Data 6). This extremely high overlap between the genes regulated by Fmp1 and Cpk1 strongly suggests that they function in the same signaling pathway during α-**a** mating. This idea was further validated by in vitro pull-down assays that verified a physical interaction between purified Cpk1 and truncated Fmp1 (735–1690 aa), which contains PH, RING and vWA domains (Fig. 4i, j). Overall, our findings support a model in which Fmp1 serves as a previously unidentified component of the *Cryptococcus* mating MAPK pathway and in which its homologs are critically involved in opposite-sex mating in evolutionarily distant fungi.

## Cqs2 bridges autodiploidization and differentiation stages, ensuring the developmental continuity of the unisexual life cycle

*Cryptococcus* unisexual reproduction is defined by pre-meiotic autodiploidization. This process is thought to be achieved via non-canonical cell cycle control, as proteins responsible for cell cycle progression have been found to be involved[36]. However, to date, the transcription factors responsible for initiation of unisexual autodiploidization have yet to be determined. To quantitatively evaluate this process, we used a previously described approach in which blastospores produced along hyphae from different strains were dissected by micromanipulation and subjected to a fluorescence-activated cell sorting (FACS)-based assay to assess their ploidy status[35,36]. With this method, we first confirmed that unisex is largely independent of the α sex-determination factors, because a mutant devoid of them underwent normal autodiploidization (Supplementary Fig. 4a). Unlike the sex-determination mutant, we identified 49 TFs affecting autodiploidization, of which Nrg1, Ada2, Hcm1, Nhp6, Zfc4, Clr1, and Cqs2 were strictly required for this process (Figs. 1d and 3a). Among these autodiploidization TFs, we focused on Cqs2, because it was dispensable for the bisex-specific process, α-**a** cell-cell fusion ($P = 0.33$, Student's $t$ test; Figs. 3a and 5a and Supplementary Fig. 4b). It has been shown that Cqs2 is the principle regulator of the oligopeptide Qsp1-directed quorum sensing system in different *Cryptococcus* pathogens[46,79,80]. Likewise, the mutant lacking Qsp1 failed to undergo autodiploidization, and this defect was partially restored by addition of a synthetic Qsp1 peptide (Supplementary Fig. 4a). These data suggest that the Qsp1-Cqs2 quorum sensing system is critical for the activation of unisexual autodiploidization.

Our functional analysis of the *cqs2Δ* mutant indicated that in addition to unisexual PD, Cqs2 engages in all cellular differentiation phases (Fig. 1d). This finding raises the possibility that Cqs2 may function in linking autodiploidization and sequential differentiation stages, ensuring the developmental continuity of the unisexual life cycle. To test this idea, we constructed a *CQS2* overexpression strain (the P*~H3~*-*CQS2* strain). Phenotypic assays indicated that overexpression of this gene remarkably promoted filamentation and basidial maturation (Fig. 5b and Supplementary Fig. 4c). In addition, imaging of mCherry-tagged Dmc1, an indicator protein for meiosis[45], showed a more abundant expression of Dmc1 in the P*~H3~*-*CQS2* strain compared with the parent strain (Fig. 5c), suggesting that overexpression of *CQS2* promoted meiotic activity. Furthermore, augmenting these pre-sporulation processes in response to overexpression led to an

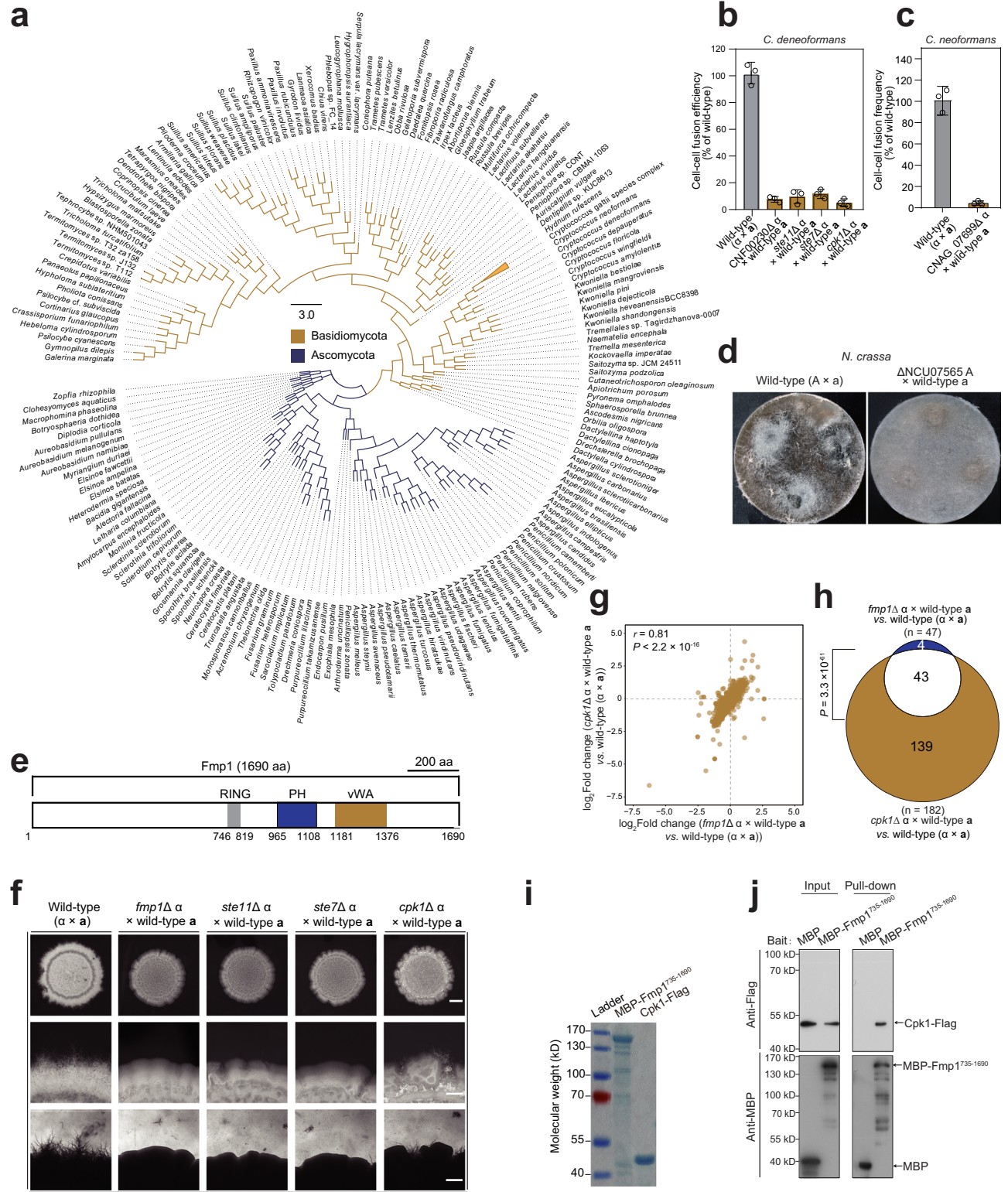

extremely robust unisexual sporulation (Fig. 5d, e). These data indicate that the intact multistage unisexual cycle can be properly promoted and coordinated in response to the increased expression of Cqs2.

Recent studies indicated that the robust unisexual reproduction observed in the XL280α strain may be attributable to the *RIC8*[G1958T] loss-of-function mutation[81,82]. To evaluate the influence of this mutation on the function of Cqs2, we overexpressed *CQS2* in another *C. deneoformans* strain, JEC21α, which does not harbor the *RIC8*[G1958T] allele[81]. Phenotypic assays indicated that overexpression of *CQS2*

substantially enhanced the robustness of filamentation compared to the wild-type strain (Supplementary Fig. 4e). Significantly, the JEC21α-derived *CQS2* overexpression strain produced robust sporulated basidia after 2 weeks of incubation on V8 agar, while sporulation was undetected in its parent strain (Supplementary Fig. 4e-f). The considerable promotion of sporulation in the overexpression strain suggested that Cqs2 is important in the induction and orchestration of the sequential unisexual stages, including pre-meiotic autodiploidization, in the JEC21α background. Taken together, these results suggest that

**Fig. 4 | Fmp1 likely serves as a component of the *Cryptococcus* mating MAPK cascade and its homologs govern opposite-sex mating in divergent fungal species. a** Phylogenetic tree of Fmp1 homologs. Protein sequences were aligned using the maximum-likelihood method with RAxML-ng v0.9. **b** Unilateral cell–cell fusion frequency of different mutants compared to the wild-type strain in *C. deneoformans*. The data are presented as the mean ± SD from three independent experiments. **c** Unilateral cell-cell fusion frequency of a CNAG_07699 deletion mutant compared with a wild-type strain in *C. neoformans*. The data are presented as the mean ± SD from three independent experiments. **d** Mating plates of the wild-type (A × a) and ΔNCU07565 mutant (ΔNCU07565 × wild-type a) were examined at 10 days post fertilization. Images are representative of three independent experiments conducted with similar results. **e** Domain organization of Fmp1. **f** Filamentation assay of unilateral crosses during bisexual mating involving a mutant strain and a wild-type mating partner XL280**a**. Cells were cultured on V8 medium for 1 week. Images are representative of three independent experiments conducted with similar results. Scale bars, 1 mm (upper panel), and 200 μm (middle and bottom panels). **g** Pearson correlation scatter plot of gene expression levels between *fmp1Δ* and *cpk1Δ* during unilateral mating revealed by RNA-seq. Two-sided Pearson test. **h**. Venn diagram indicating the overlap between the Fmp1-controlled genes and the Cpk1-controlled genes. The *P* value was determined with one-sided Fisher's exact test. **i**. Coomassie stained SDS gel showing the purified proteins used for pull-down assays. Truncated Fmp1 (735–1690 aa) was fused with maltose binding protein (MBP) in the N-terminus All proteins were expressed in Rosetta *E. coli* and purified via affinity chromatography using Ni-NTA resin. Images are representative of two independent experiments conducted with similar results. **j**. MBP pull-down assays to determine the interaction between MBP-Fmp1[735–1690] and Cpk1-FLAG. Purified MBP-Fmp1[735–1690] or MBP (control) proteins were used to pull down Cpk1-FLAG using MBP-agarose beads. Anti-FLAG and anti-MBP antibodies were used for the immunoblotting. Images are representative of three independent experiments conducted with similar results. Source data are provided as a Source Data file.

the key role of Cqs2 in promotion and coordination of the multistage unisexual cycle is not unique to XL280α but is likely conserved in different *C. deneoformans* isolates.

## Pum1, as a key target of Cqs2, activates unisexual autodiploidization

To explore the molecular mechanisms underlying the actions of Cqs2 during unisexual reproduction, we employed two independent chromatin immunoprecipitation sequencing (ChIP-seq) experiments. The two ChIP-seq approaches co-identified significant overlaps in the binding events (Fig. 5f, g and Supplementary Data 7), which were further analyzed to determine the direct regulon of Cqs2 and its binding motif during unisexual reproduction (Fig. 5h). Based on the ChIP-seq analysis, Cqs2 was found not to bind directly to the promoter regions of any sex-determination genes, and no remarkable changes in the expression of these genes were detected in its absence (Fig. 5i). It should be noted that compared with the wild-type strain, there was an increase of approximately 2-fold in the mRNA levels of the *MFα* gene in the *CQS2*-overexpressing strain after 12 h of unisexual induction, and similar increases in the mRNA levels of the *STE3α* and *SIX1α* genes were observed 24 h after induction (Fig. 5i). The asynchronous induction of these genes may be due to an indirect effect of the forced expression of Cqs2, even though it was not adequate to enable the *CQS2*-overexpressing strain to undergo syngamy, considering the unaltered α-**a** cell mating efficiency in this strain (Supplementary Fig. 4d).

We found that the direct regulon of Cqs2 contained multiple genes involved in various stages of the unisexual cycle (Fig. 5i). This breadth of action is consistent with the coordinating role of Cqs2 in the unisexual cycle (Fig. 5a–e). Among these genes, we focused on *PUM1*, which encodes a Pumilio family RNA-binding protein[45]. Pum1, like its upstream regulator Cqs2, has been reported to have roles in multiple stages during unisexual reproduction, but not in bisexual syngamy[45], suggesting that Pum1 may be an important target of Cqs2 during the unisexual cycle. This notion was further supported by the following findings. First, ChIP-seq and qRT-PCR assays indicated that Cqs2 directly stimulated the expression of *PUM1* during unisexual reproduction (Fig. 5j, k). Second, disruption of *PUM1* led to severe defects in unisexual autodiploidization (Fig. 5a). Third, the Pum1 regulon was found to include 45.4% of Cqs2 targets (Fig. 5l and Supplementary Data 8). Finally, overexpression of *CQS2* in the *pum1Δ* mutant failed to restore the defects in unisexual filamentation, basidial maturation, and sporulation (Fig. 6a, b).

To further explore the correlation between *PUM1* expression and autodiploidization, we constructed an inducible expression system of *PUM1* (P*CTR4-2*-*PUM1-mCHERRY*α), which harbors a construct directing the expression of mCherry-tagged Pum1 under the control of the inducible copper transporter *CTR4* promoter (copper deprivation–on; copper repletion–off)[83] (Supplementary Fig. 5a, b). We found that once

*PUM1* was induced, even under sex-suppressing conditions, considerable increases in cell size and in the population of the diploid cells were detected (Fig. 6c, d). Of note, the diploid cells produced by inducing *PUM1* expression retained a yeast morphology (Supplementary Fig. 5c). These results demonstrate that enhanced expression of *PUM1* is sufficient to drive autodiploidization regardless of sex-inducing cues and developmental context. Pum1 overexpression was also found to efficiently stimulate autodiploidization in the absence of Cqs2, but not vice versa (Supplementary Fig. 5d, e), suggesting that Pum1 may function as a terminal regulator of autodiploidization. We also constructed a JEC21α-derived P*CTR4-2*-*PUM1-mCHERRY* strain to test whether inducing the expression of *PUM1* can similarly promote autodiploidization in this background. As expected, enhanced expression of Pum1 resulted in a remarkable increase in cell size and a prominent population of diploid cells, as determined by a FACS-based ploidy evaluation (Supplementary Fig. 5f-g).

## Pum1-mediated autodiploidization requires *CUA1*, which encodes a spindle pole body-associated protein

To test whether induction of *PUM1* can likewise stimulate ploidy increase in diploid cells, we introduced the inducible expression system of *PUM1* into the XL280α/**a** background to generate a P*CTR4-2*-*PUM1-mCHERRY*α/**a** strain (Fig. 6e). FACS-based ploidy analysis indicated that overexpression of *PUM1* in diploid α/**a** cells did not change ploidy (Fig. 6f), suggesting that Pum1-stimulated ploidy doubling is dependent on haploid status.

We next sought to understand the mechanism underlying the selective activation of ploidy doubling in haploid cells. Considering that ploidy change was shown to be attributed to cell cycle control, we concentrated on 13 targets that were shown by RNA-seq analysis to be upregulated by both Pum1 and Cqs2. These targets contained at least one domain predicted by the Pfam program to be related to cell cycle processes (Supplementary Data 9). Of these targets, we found the cyclin *CBC1* (also known as *CLB3*)[84], which has been reported to affect autodiploidization in XL280α[36], highlighting the potential involvement of other target genes in this process.

To measure the effect of ploidy status on Pum1-mediated transcriptional induction of these targets, we conducted qRT-PCR assays to measure the production of the relevant mRNAs in XL280α, XL280α/**a**, P*CTR4-2*-*PUM1-mCHERRY*α and P*CTR4-2*-*PUM1-mCHERRY*α/**a** strains that were cultured under mating-inducing condition (Fig. 7a and Supplementary Fig. 6). It was shown that the strength of the Pum1-mediated induction of the target genes in diploid α/**a** cells was markedly lower than in haploid cells (Fig. 7a), and more than half (7 genes) of the 13 cell cycle-related genes showed only a marginal induction (fold change <2), or even non-induction, in α/**a** cells with forced Pum1 expression (Fig. 7a). These data suggest that the failure to induce ploidy change in α/**a** cells by Pum1 may be attributed to its inability to sufficiently

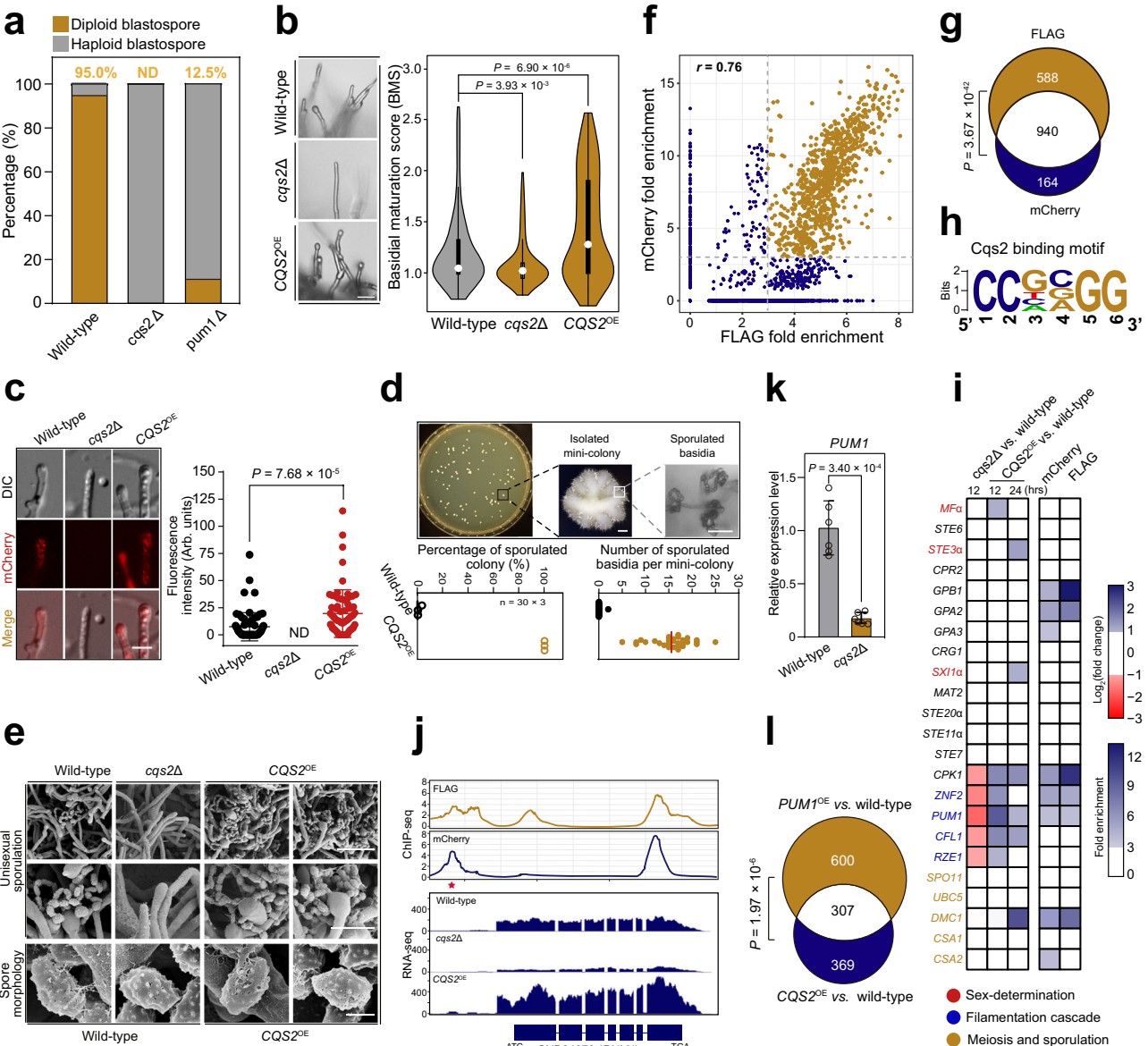

**Fig. 5 | Cqs2 drives and coordinates multistage unisexual reproduction by coordinating the expression of multiple genes involved in various unisexual stages. a** FACS-based ploidy assessment of blastospores dissected from unisexual hyphae of different strains. For each strain, 40 blastospores dissected from different unisexual hyphae were tested for ploidy level. ND not detected. **b** Basidial formation and BMS distribution of wild-type, *cqs2Δ*, and *CQS2*^OE^ (P_H3-CQS2). Hyphae with or without basidia were photographed at 5 days post unisexual stimulation and were randomly chosen for the BMS calculation (*n* =150 for each strain). Boxplots show the median and the upper and lower central quartiles. The expected range of the data is indicated by whiskers. Scale bar: 10 μm. Two-tailed Student's *t* test. **c** The fluorescence-labeled strains were photographed at 3 days post-inoculation on V8 agar to stimulate unisexual reproduction. For each strain, 70 basidia were examined for the expression of Dmc1-mCherry. Data are presented as the mean ± SD. ND not detected. Two-tailed Student's *t* test; scale bar: 10 μm. **d** Diagram showing the approach for quantitative analysis of unisexual sporulation. Scale bars: 200 μm (upper middle panel) and 20 μm (upper right panel). **e** Sporulation phenotypes of wild-type, *cqs2Δ* and *CQS2*^OE^. The *CQS2*^OE^ strain produced ample basidia with four chains of morphologically regular basidiospores

during the unisexual cycle. Images are representative of three independent experiments conducted with similar results. Scale bar, 10 μm (upper panel), 5 μm (middle panel), and 1 μm (bottom panel). **f** Binding events identified by FLAG and mCherry tagged ChIP-seq approaches. Pearson correlation coefficient (*r*). **g** Significant overlap between binding events identified by two ChIP-seq approaches. One-sided Fisher's exact test. **h** The binding motif of Cqs2. **i** Cqs2 directly activated genes involved in various stages of sexual development, but did not bind to the promoters of the sex-determination genes. **j** Genome browser images depicting ChIP-seq enrichment (upper panels) and relative transcript levels based on RNA-seq (bottom panels) at the *PUM1* locus in different strains. **k** qRT-PCR showing the mRNA level of *PUM1* in *cqs2Δ* during unisexual reproduction compared to wild-type. Data are presented as the mean ± SD of six independent experiments, two-tailed Student's *t* test. **l** The overlap between *CQS2*-regulated genes revealed by RNA-seq analysis of the *CQS2* overexpression strain and *PUM1*-controlled genes, which were identified based on the published RNA-seq data targeting the *PUM1*^OE^ strain[21]. One-sided Fisher's exact test. Source data are provided as a Source Data file.

increase the expression of these putative cell cycle-related genes in a coordinated manner in diploid cells.

To confirm this hypothesis, we attempted to create deletion strains for each of seven genes that were selectively regulated by Pum1

in the haploid context. As a result, the deletion mutants of five genes were obtained, whereas repeated attempts to knock out the other two genes failed, suggesting that they may be essential genes. We then performed FACS-based ploidy analysis on unisexual blastospores

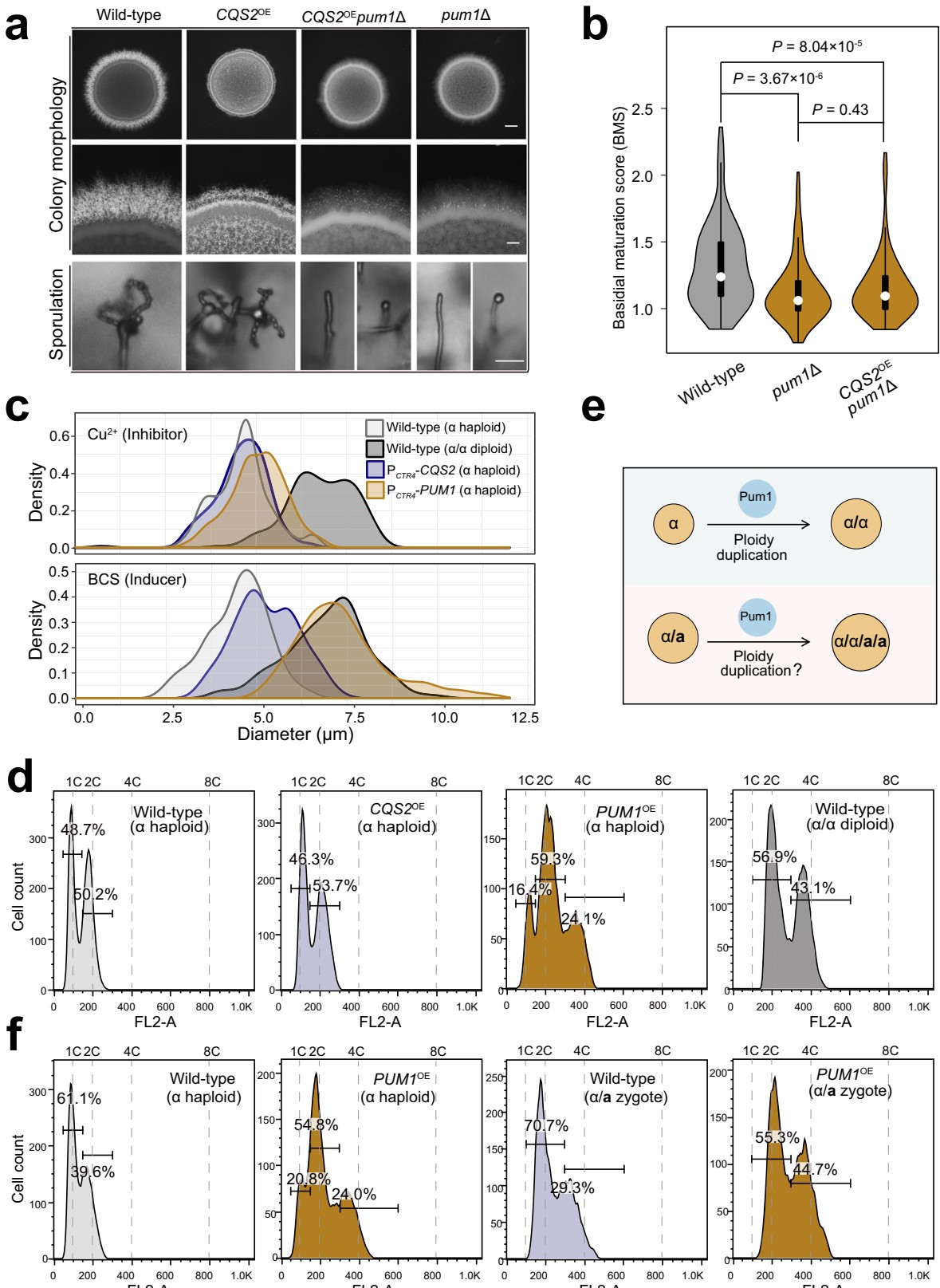

isolated from the obtained deletion strains. Of them, disruption of CNA06220 was found to almost completely abolish the cryptococcal ability to produce diploid blastospores (Fig. 7b). Moreover, Pum1 overexpression was not able to induce autodiploidization in the absence of CNA06220. These results suggest that CNA06220 plays an important role in Pum1-mediated autodiploidization, and we named

this gene *Cryptococcus* unisexual autodiploidization 1 (*CUA1*; Fig. 7c). We further showed that the phenotypes of the *cua1*Δ mutant are highly similar to those of the *pum1*Δ strain; both of these mutant strains produced short hyphae and were not able to give rise to four chains of basidiospores (Fig. 7d). In addition, like Pum1, Cua1 appears not to affect α-**a** cell–cell fusion (Fig. 7e). Furthermore, a comprehensive

**Fig. 6 | Pum1 is a key target of Cqs2 during unisexual autodiploidization. a.** The colony morphology, self-filamentation and sporulation phenotypes of various mutants. Hyphae and chains of basidiospores were photographed after 7 days of growth on V8 medium. Images are representative of three independent experiments conducted with similar results. Scale bars, 1 mm (upper and middle panel) and 20 μm (bottom panel). **b.** Overexpression of *CQS2* did not restore the defect of basidial development in the *pum1Δ* mutant. Strains were cultured on V8 agar for 7 days, and 100 hyphae of each strain were randomly chosen for BMS calculation. Two-tailed Student's *t*-test. **c.** Cell size distribution of wild-type, $P_{CTR4}$-*PUM1*, and $P_{CTR4}$-*CQS2* cultured under sex-suppressing conditions (YPD liquid medium). Strains were cultured in YPD liquid medium for 3 days in the presence of copper (inhibitor of the *CTR4* promoter, final concentration 25 μM) or BCS (inducer of the *CTR4* promoter, final concentration 20 μM) and over 150 cells of each strain were examined for their diameters. **d.** FACS-based ploidy level assessment of wild-typeα, $P_{CTR4}$-*PUM1*α, $P_{CTR4}$-*CQS2*α and wild-typeα/α cultured under sex-suppressing condition. Strains were cultured in YPD liquid medium for 3 days in the presence of BCS (inducer of *CTR4* promoter, final concentration 20 μM), and the ploidy of each strain was subsequently assessed by FACS analysis. The gating strategy is shown in Supplementary Fig. 5d. **e.** Diagram depicting the roles of Pum1 in stimulating ploidy increase in α haploids or α/**a** zygotes. **f.** FACS-based ploidy assessment of wild-typeα, *PUM1*<sup>OE</sup>α, wild-typeα/**a**, and *PUM1*<sup>OE</sup>α/**a**. Strains were cultured in YPD liquid medium for 3 days in the presence of BCS (final concentration 20 μM) before FACS analysis. The gating strategy is shown in Supplementary Fig. 5d. Source data are provided as a Source Data file.

phenotypic analysis together with a murine model of infection indicated a wild-type-like growth of *cua1Δ* and *pum1Δ* under 27 distinct in vitro or in vivo conditions, suggesting that they are specific for unisexual autodiploidization and development (Supplementary Fig. S7).

Cua1 contains 3436 amino acids and has a relative molecular weight of approximately 365.6 kD, making it one of the largest proteins produced in *C. deneoformans*. Phylogenetic analyses showed that the homologs of Cua1 are distributed almost exclusively in species from *Cryptococcus* and its sister genus *Kwoniella*. Functional domain prediction indicated that this protein contains three putative domains (PHA03307, PHA03247 and TolA) (Fig. 7f). Interestingly, PHA03307 and PHA03247 domains are mostly found in virus proteins. In particular, the PHA03307-like domain from virus nuclear protein ICP4 has been reported to be required for interference with host cell cycle progress[85], suggesting a possible functional role of Cua1 in cell cycle control.

To further explore the molecular function of this protein, we constructed a strain harboring a gene leading to constitutive production of mCherry-fused Cua1. As shown in Fig. 7g, the Cua1-mCherry protein was exclusively localized in foci in proximity to the nucleus in various phases of the cell cycle. This localization pattern is similar to that of the components of the spindle pole body (SPB), which have been described to be tethered to the nucleus during cell cycle progression in budding yeast[86]. Time-lapse imaging of Cua1-mCherry and GFP-tagged Spc98, a molecular indicator of the *Cryptococcus* SPB[87,88], revealed co-localization throughout the cell cycle (Fig. 7h). These data indicate that Cua1 is a previously unidentified SPB-associated protein that is critical for the Pum1-mediated unisexual autodiploidization.

## Discussion

Even though sexual reproduction is a universal characteristic of nearly all eukaryotic organisms, diverse sexual forms have been adopted by various eukaryotes. For instance, it is well known that the fusion of gametes (known as syngamy or fertilization) with identical ploidies followed by nuclear fusion enables an overall doubling of ploidy prior to meiosis[26,35,71,89,90]. In addition to syngamy-dependent ploidy changes, evolutionarily divergent eukaryotes utilize autodiploidization or autopolyploidization mechanisms to accomplish a pre-meiotic ploidy increase during syngamy-independent sexual (meiotic) reproduction[30,35,36,91,92].

The basidiomycete *C. deneoformans* possesses both syngamy-dependent and autodiploidization-dependent sexual life cycles, which may provide reproductive flexibility for maximizing its survival in unpredictable environments[11,13,28,81,82,93,94]. In the Basidiomycota, syngamy is typically determined by an ancient conserved sex-determination circuit, which generally consists of a homeobox family regulator, pheromone, and pheromone receptor[26,65,95]. Before syngamy, the recognition between a pheromone and its cognate receptor is the key compatibility checkpoint. Once mating partners with compatible pheromone and cognate receptor pairs have fused, homeobox family regulators regulate a transcriptional cascade that promotes further sexual development[95–97]. In *C. deneoformans*, the α sex-

determination factors are critical for bisexual syngamy, while they are largely dispensable for the unisexual autodiploidization of α cells. This dispensability is likely due to the incompatibility of the pheromone (Mfα) and the pheromone receptor (Ste3α) produced by α cells, which may prevent the co-occurrence of syngamy and autodiploidization, thus avoiding unfit polyploidy during α unisexual reproduction.

Unlike the α sex-determination factors, we found that the Qsp1-Cqs2-Pum1 regulatory cascade is essential for unisexual autodiploidization. In this cascade, Pum1 likely acts as a terminal activator for initiating autodiploidization, because its overexpression is sufficient to trigger ploidy doubling in haploid cells regardless of sex-inducing cues and developmental context. Interestingly, the forced expression of Pum1 did not elicit ploidy duplication in diploid α/**a** cells. One plausible mechanism for the inability of Pum1 to change ploidy in diploid cells may stem from a dilution of the intracellular concentrations of the elements associated with the function of Pum1 in diploid cells due to their larger cellular volume. This dilution would be expected to result in activity levels that are insufficient to induce the expression of cell cycle genes and, as a result, a failure to increase ploidy. Such a cell size-dependent protein dilution strategy has also been reported in other yeast species, in which it is required for the balance between cell cycle and cell growth[98,99]. The inability of Pum1 to increase ploidy in diploid cells likely benefits ploidy maintenance after sexual diploidization and helps to avoid the occurrence of polyploidy, which could negatively impact sexual development and meiotic progression.

We identified an unusually large protein, Cua1, that is essential for Pum1-triggered unisexual PD but that does not affect bisexual cell-cell fusion. Monitoring of the growth of a mutant devoid of Cua1 under 27 in vitro and in vivo conditions highlights its functional specificity in unisexual autodiploidization and development. We showed that Cua1 acts as a SPB-associated protein, according to its co-localization with an indicator protein of *Cryptococcus* SPB. Interestingly, accumulating studies in both *S. cerevisiae* and human cells have indicated that disturbing SPB assembly serves as an important mechanism for ploidy increase[100–103]. For example, in *S. cerevisiae*, a dysfunctional SPB is capable of nucleating microtubules from the cytoplasmic side but not the nuclear side, leading to a non-disjunction event and spontaneous diploidization[100]. Further investigation into the role of Cua1 in connecting SPB activity and autodiploidization may lead to a deepened understanding of the mechanisms that trigger this unisex-specific process.

Phylogenetic analyses indicated that *CUA1* orthologues nearly exclusively exist in *Cryptococcaceae* species. Notably, the key components of the Qsp1-directed quorum sensing system, which functions upstream of *CUA1* during unisexual reproduction, have been shown to be distributed in species belonging to the same taxonomic group[104], leading to the hypothesis that this system was acquired recently. In comparison, the Pumilio family RNA-binding protein Pum1, which is also known as Puf3 in some fungal species, as well as its RNA binding motif, are conserved across the fungal kingdom[105,106]. Despite this

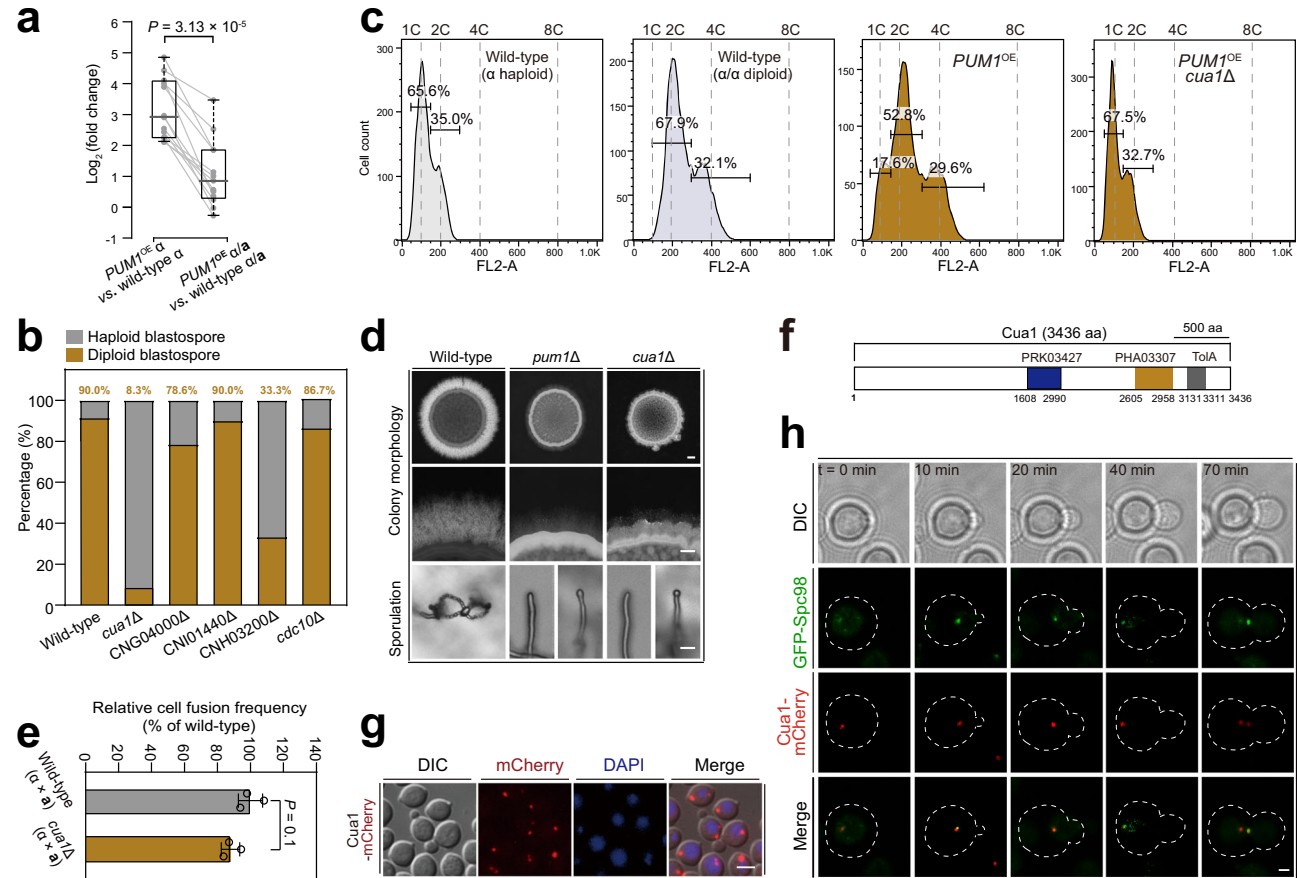

**Fig. 7 | *CUA1* is essential for Pum1-stimulated ploidy doubling and unisexual reproduction. a** Relative mRNA levels of the cell cycle genes in different strains as determined by qRT-PCR. Each dot indicates the mean of five independent experiments, two-tailed Student's *t* test. Boxplots show the median and the upper and lower central quartiles. The expected range of the data is indicated by whiskers. **b** FACS-based ploidy assessment of blastospores dissected from unisexual hyphae of different strains. For each strain, over 10 blastospores dissected from different unisexual hyphae were tested for ploidy level. The gating strategy is shown in Supplementary Fig. 5d. **c** FACS-based ploidy level assessment of different strains cultured under sex-suppressing condition. Strains were cultured in YPD liquid medium for 3 days in the presence of BCS, an inducer of the *CTR4* promoter (20 μM), and the ploidy of each strain was subsequently assessed by FACS analysis. **d** The colony morphology, self-filamentation and sporulation phenotypes of

different mutants. Hyphae and chains of basidiospores were photographed after 7 days of growth on V8 medium. Images are representative of three independent experiments conducted with similar results. Scale bars, 1 mm (upper and middle panel) and 20 μm (bottom panel). **e** Bilateral *cua1* mutant cell-cell fusion frequency compared to the wild-type strain. The data are presented as the mean ± SD from three independent experiments, two-tailed Student's *t* test. **f** Domain organization of Cua1. **g** Subcellular localization of Cua1-mCherry. DAPI (4′,6-diamidino-2-phenylindole) was used to visualize nuclei. Images are representative of three independent experiments conducted with similar results. Scale bar, 5 μm. **h** Cells expressing Cua1-mCherry and a marker of the spindle pole body, GFP-Spc98, were analyzed by time-lapse microscopy. Numbers indicate times in minutes. Images are representative of three independent experiments conducted with similar results. Bar = 1 μm. Source data are provided as a Source Data file.

conservation, it has been suggested that rewiring of Pum1-orchestrated mRNA targets occurred at different points in evolution, potentially contributing to biological diversification[107]. Thus, the Qsp1-Cqs2-Pum1-Cua1 genetic circuit likely evolved after the emergence of *Cryptococcaceae* to be co-opted for unisexual reproduction, supporting a previous hypothesis that unisex appears to be a relatively recent evolutionary innovation[27]. This evolutionary innovation conferred the abilities of C. *deneoformans* and other important *Cryptococcus* pathogens with a marked predominance of only one mating type to reproduce unisexually, which may have helped them to become successful global fungal pathogens.

## Methods

### Media and growth conditions

Yeast cells were grown routinely at 30 °C on YPD (1% yeast extract, 2% Bacto peptone, 2% dextrose, and 2% Bacto agar) medium. Sexual differentiation and bisexual cell−cell fusion assays were conducted on V8 agar (0.5 g/liter $KH_2PO_4$, 4% Bacto agar and 5% V8 juice, pH = 7.0) at 25 °C in the dark for the designated time period. For dissection of

blastospores, strains were cultured on Murashige and Skoog (MS) medium minus sucrose (Sigma-Aldrich) for 2 weeks at 25 °C in the dark.

### Construction of the *C. deneoformans* TF mutant library

TF mutant strains were constructed in the *C. deneoformans* XL280α strain background. Targeted gene deletion was performed as described previously[44]. Briefly, a deletion construct consisting of approximately1 kb of sequence flanking the target gene and splitting part of the neomycin (*NEO*) resistance cassette was introduced into relevant XL280α cells by biolistic transformation. All of the primers used in this study are listed in Supplementary Data 10.

To generate the deletion constructs, the 5′ and 3′ flanking regions of the targeted transcription factor were amplified with primer pairs p1/p2 and p3/p4 with XL280α genomic DNA as template. The resistance cassette for the selectable marker *NEO* was amplified with primer pair M13F/ M13R from plasmid pPZP-NEO1[108]. The three products were then fused together by a triple-joint overlap PCR step. In the third round of PCR, the p5 primer in the inner part of 5′ flanking region and the p6 primer in the *NEO* cassette were used to generate a 5′ flanking

sequence connected with *NEO* split marker (5′-*NEO*). Similarly, the 3′ flanking region was connected with *NEO* split marker (3′-*NEO*) by using primer pair p7/p8. Transformants were selected on YPD agar containing G418. All G418 resistant strains were confirmed by diagnostic PCR. To validate a mutant phenotype and to exclude any unlinked mutational effects, at least two independent deletion strains for each TF were constructed. All of the strains used in this study are listed in Supplementary Data 11.

## Phenotypic analysis

All phenotypic traits were evaluated based on at least two independent TF mutants. When two independent TF mutants showed inconsistent phenotypes, additional TF mutants were assessed. Quantitative analysis of hyphal initiation was conducted as previously described[46]. Briefly, low-density cells were dropped onto V8 agar to allow the formation of isolated mini-colonies after approximately 21 h of cultivation. The mini-colonies exhibited heterogeneity in filamentation, which is indicative of the strength of unisexual induction among mini-colonies. For each strain, 100 mini-colonies were randomly selected for calculating of filamentous frequency (FF). The relative FF value of each mutant was normalized by dividing by the wild-type FF value (mutant FF/wild-type FF).

Hyphal extension, aerial structures, and invasive structures were evaluated quantitatively or semi-quantitatively as follows: 3 μL of cells ($A_{600} = 1.0$) were spotted onto V8 agar and then incubated for 7 days at 25 °C in the dark. Under these conditions, ample hyphae were typically formed on the edges of the colonies. The filamentous cells also extended into the air (aerial structures) or invaded into the medium (invasive structures). Colony morphologies were visualized with a Zeiss Microscope, and the lengths of hyphae on the edge the colony were measured using Zen 2016 software. For each strain, 100 hyphae along the colony edges were randomly selected and measured for length. The relative hyphal length of each mutant strain was normalized by dividing the length of mutant hyphae by the length of wild-type hyphae. Aerial structures were evaluated based on the robustness of the white fluffy appearance on the surfaces of the colonies. Invasive structures were assessed according to the amount of cells that remained when the surface was washed.

Basidial maturation was quantified by the basidial maturation score (BMS), which was assessed as previously described[21]. The relative BMS of each mutant strain was normalized by dividing the BMS of the mutant strain by the BMS of the wild-type strain.

For quantitative analyses of sporulation, the cells of different strains were plated onto MS medium at a low cell density to form isolated mini-colonies. After 7 days of culturing, the aerial hyphae in the mini-colonies showed heterogeneity in differentiation into sporulated basidia. The sporulation incidence among the hyphae from the mini-colonies positively correlates with the strength of unisexual sporulation. For each mutant, 5 mini-colonies were randomly selected, and the total amount of sporulated basidia were counted. The relative sporulation value for each mutant strain was normalized by dividing the mutant sporulation value by the wild-type sporulation value.

Autodiploidization (unisexual PD) was reflected by the percentage of diploid blastospores dissected from the unisexual hyphae of each strain. Blastospore dissection and FACS-based evaluation of ploidy level were performed as previously described[35]. The ploidy of the blastospores was determined by FACS analysis as previously described[109,110]. A total of 10,000 cells were analyzed for each sample, using the FL2 channel on a BD FACSCalibur™ flow cytometer at the Beijing Regional Center of Life Science Instrument, Chinese Academy of Science. FlowJo (v10.0) software was used to perform data analysis.

Cell-cell fusion (bisexual PD) assays were performed as previously described[56]. The relative cell-cell fusion efficiency for each mutant strain was normalized by dividing the mutant cell-cell fusion efficiency by the wild-type cell-cell fusion efficiency.

## Gene overexpression

For *CQS2* overexpression, a promoter swapping technique was used to place the coding-region of *CQS2* downstream of the promoter of the gene encoding histone H3. The expression of this gene was determined to be abundant and steady throughout unisexual development, based on the reanalysis of previous RNA-seq data[21]. For the inducible expression of *PUM1* and *CQS2*, the overexpression constructs were created by amplifying the entire ORFs and inserting them into plasmid pXC-mCherry after the *CTR4* promoter as previously described[44]. The overexpression constructs were linearized and introduced into relevant *Cryptococcus* strains by electroporation. The primers used for the generation of the overexpression strains are listed in Supplementary Data 11.

## Murine model of cryptococcosis

Yeast strains were grown in YPD liquid medium at 30 °C overnight, and the cells were collected and washed twice with sterile PBS buffer and diluted to a final concentration of $1 \times 10^6$ CFU mL$^{-1}$. C57BL/6 mice were purchased from Vital River (Beijing, China). The mice were routinely maintained in a pathogen-free animal facility at a temperature of 21 °C, relative humidity of 50–70%. Mice were cared with an alternating 12 h light-dark cycle and unlimited food and water supply. Female mice (7–8 weeks old) were subjected to cryptococcal infection by intravenous tail vein injection. For survival assays, mice ($n = 10$) were randomly distributed into each group. The mice were infected with 100 μL of fungal cell suspensions via intranasal instillation. After infection, animals were observed daily for disease progression.

## Organ fungal burden analysis

The infection procedure for organ fungal burden analyses was the same as for the survival assay. The brains of three infected mice at 7 days post infection were homogenized and spread on a YPD plate containing 100 μg/mL of chloramphenicol, and colonies were counted after 2–3 days of incubation at 30 °C.

## Microscopy and fluorescence

To examine the expression and subcellular localization of Dmc1, strains harboring P$_{DMC1}$-*DMC1*-*mCherry* were spotted onto V8 agar at 25 °C in the dark for a designated period of time. Images were taken with a Zeiss Axioplan 2 imaging system, and images were analyzed with the AxioCam MRm camera software Zen 2016 (Carl Zeiss Microscopy). For live-cell imaging, poly-L-lysine (1 μg/mL) was used to coat the bottom center of Nunc glass-bottom dishes, and the resulting poly-L-lysine-coated dishes were used for the trapping of living fungal cells for microscopic examination.

Strains harboring P$_{CTR4}$-*CUA1*-*mCherry* and P$_{UGE2}$-*GFP*-SPC*98* were cultured in a poly-L-lysine-coated dish containing YPG medium supplemented with the copper chelator bathocuproine disulfonate (BCS; 40 mM). The dish was then washed twice with PBS to remove unattached fungal cells. Fresh YPG medium supplemented with 40 μM BCS was added, and the subcellular localizations of Cua1-mCherry and GFP-Spc98 were visualized using ultrahigh-resolution laser confocal microscopy (Nikon A1-N-SIM S). The excitation filters used were 475/28 for GFP and 575/25 for mCherry. The emission filters were 525/50 for GFP and 632/60 for mCherry. The cells were photographed every 10 min for 4 h. Images were processed using the Zeiss image processing software Zen 2016 (Carl Zeiss Microscopy), Nikon NIS-Elements AR Ver5.40.00 (Nikon Corporation), or Photoshop CS6 (Adobe Systems, San Jose, CA).

## *N. crassa* fertility testing

The *N. crassa* wild-type and mutant strains were obtained from the *N. crassa* deletion collection at the Fungal Genetics Stock Center (FGSC)[73], and knockout event of the mutant was confirmed by PCR. Tests of the fertility of *N. crassa* were performed using standard

techniques[111]. Strains were grown on Vogel's minimal media (VM) in slant tubes for 1 week to induce conidiation. For crossing, *mat A* strains cultured on synthetic crossing medium (SCM) plates for 7 days were fertilized with conidia of *mat a* strains and further incubated for 7 days to induce the formation of perithecia.

## DAPI staining

DAPI (4′,6-diamidino-2-phenylindole) staining assays were performed as previously described[112]. Briefly, yeast cells or hyphae were collected and fixed with 3.7% formaldehyde and permeabilized in 1% Triton X-100. The cells were then washed three times with PBS and incubated in 2 µg/mL DAPI before being dropped onto a glass slide for fluorescent microscopic observation.

## Scanning electron microscopy

Scanning electron microscopy (SEM) was performed at the Beijing Regional Center of Life Science Instrument, Chinese Academy of Science. Samples were prepared as previously described[35]. The cells were spotted onto V8 solid medium and then incubated at 25 °C in the dark for 7 days. Unisexual colony patches were then excised and fixed in phosphate-buffered glutaraldehyde (pH 7.2) at 4 °C for at least 2 h. The fixed samples were rinsed in deionized distilled water (ddH$_2$O) three times for 6 min, 7 min and 8 min. The samples were dehydrated in a graded ethanol series (50%, 70%, 85%, and 95%) by incubating for 14 min at each concentration and then dehydrated with incubation in 100% ethanol three times for 15 min each. After dehydration, the blocks were critical-point dried with liquid CO$_2$ (Leica EM CPD300, Germany) and sputter coated with gold-palladium (E-1045 ion sputter, Hitachi, Japan). The samples were viewed and images were taken using a Quanta200 scanning electron microscope (FEI, USA).

## RNA purification and quantitative RT-PCR analysis

RNA extraction and quantitative real-time PCR (qRT-PCR) were carried out as described previously[46]. Briefly, the cells were frozen in liquid nitrogen and ground to a fine powder and total RNA was extracted and purified using an Ultrapure RNA Kit (Kangweishiji, CW0581S) according to the manufacturer's instructions. A Fastquant RT Kit (Tiangen KR106-02, with gDNase) was used for first-strand cDNA synthesis following the manufacturer's instructions. The qRT-PCR was conducted using power SYBR qPCR premix reagents (KAPA) in a CFX96 Touch™ Real-time PCR detection system (BioRad), and relative transcript levels were calculated as fold changes and normalized to the housekeeping gene *TEF1* using the comparative $C_t$ method as previously described[44]. Data were visualized using GraphPad Prism 8.0.

## RNA-seq and data analysis

For RNA-seq analyses, strains were cultured in YPD liquid medium at 30 °C overnight. The cells were washed with ddH$_2$O and spotted on V8 medium to stimulate unisexual reproduction. The level and integrity of RNA in each sample were evaluated using a Qubit RNA Assay Kit on a Qubit 2.0 Flurometer (Life Technologies, CA, USA) and RNA Nano 6000 Assay Kit with the Bioanalyzer 2100 system (Agilent Technologies, CA, USA), respectively. RNA purity was assessed using a Nano Photometer spectrophotometer (IMPLEN, CA, USA). The transcriptome libraries were generated using the VAHTS mRNA-seq v2 Library Prep Kit (Vazyme Biotech Co., Ltd, Nanjing, China) according to the manufacturer's instructions.

Sequencing the transcriptome libraries was performed by Annoroad Gene Technology Co., Ltd (Beijing, China). The samples were clustered using VAHRS RNA adapters set1/set2 and sequenced on an Illumina platform. For RNA-seq analysis, the quality of sequenced clean data was analyzed using FastQC v0.11.5 software. Subsequently, sequences from approximately 2 GB of clean data for each sample were mapped to the genome sequence of *C. neoformans* XL280α using STAT_2.6.0c. Gene expression levels were measured in transcripts per

million (TPM) by Stringtie v1.3.3 to determine unigenes. All unigenes were subsequently aligned against the well-annotated genome of JEC21; this strain is congenic to JEC20**a**, which served as the parent strain to generate XL280α through a cross with B3501α. The differential expression of genes (DEGs) was assessed using DEseq2 v1.16.1 of the R package and defined based on the fold change criterion (log$_2$| fold-change | > 1.0, $q$ value < 0.01).

## Chromatin immunoprecipitation sequencing and data analysis

ChIP was performed as described previously with modifications[113]. Briefly, strains expressing Cqs2-FLAG and Cqs2-mCherry were diluted and spotted onto V8 agar containing 200 µM BCS for 8 h at 25 °C in the dark. The cells were harvested and cross-linked with 1% formaldehyde at room temperature for 15 min. Glycine was then added to a final concentration of 125 mM, and the treated cultures were incubated for another 5 min at room temperature. Cell pellets were harvested by centrifugation at 4000 rpm for 5 min at 4 °C and washed three times with ice-cold water. Cell pellets were then ground to a fine powder in liquid nitrogen.

Chromatin fragmentation was performed by digestion using 2 units of microccocal nuclease for 15 min at 37 °C. The nuclear membrane was disrupted by sonication in a Bioruptor water bath sonicator (Diagenode; 3 cycles of 30 s on, 30 s off at the high setting) at 4 °C. After centrifugation at 12,000×$g$ for 10 min at 4 °C, the supernatant was collected, and a 50 µL aliquot of sample was reserved as ChIP input material. For the strain samples harboring Cqs2 tagged with mCherry or FLAG, 20 µL of RFP-TRAP magnetic beads (rtma-20, Chromo Tek, 1:200 dilution) or anti-FLAG M2 magnetic beads (M8823, Sigma-Aldrich, 1:200 dilution) were added to the remaining lysate. All samples were incubated overnight at 4 °C with agitation. The beads were washed 5 min at 4 °C with ice-cold buffers as follows: twice with lysis buffer, once with high-salt buffer, once with LiCl buffer, and once with TE buffer. After the final wash, the bound chromatin was eluted with 250 µL elution buffer. Cross-links of input and IP material were reversed by adding 10 µL 5 M NaCl and 4 µL 10 mg mL$^{-1}$ Proteinase K (Promega) and incubating for 4 h at 65 °C. DNA was subjected to phenol-chloroform extraction and ethanol precipitation. The isolated DNA was dried and resuspended in 20 µL TE containing 10 µg mL$^{-1}$ RNase.

For library construction, 5 ng of ChIP input and IP material were end-repaired with Klenow DNA polymerase, A-tailed using the Klenow fragment lacking exonuclease activity and ligated to adapters. The ligated fragments were amplified using indexed primers (Illumina) for 8 to 10 PCR cycles. DNA was purified with 1.8 volumes of Ampure XP DNA purification beads between each step.

Library sequencing was performed on an Illumina Hi-Seq 2000. The raw data were aligned using Bowtie2 (version 2.1.0)[114] with default parameters. Indexed and sorted bam files of each dataset were created using SAMtools (version 1.5)[115]. ChIP-Seq peaks were called using MACS2 (version 2.1.1.20160309)[116]. Annotation was performed by using ChIPseeker (R 1.12.1) to localize each peak to promoter regions of related genes. Chromosomal traces were plotted using Integrative Genomics Browser 9.0.0 (Broad Institute). ChIP-Seq peaks of the mCherry and FLAG tags were merged using HOMER (4.9.1), and 50 bp upstream and downstream merged peaks were selected to find conserved binding motifs using HOMER (4.9.1). The motifs were generated for widths of 6 to 14 bps for both strands represented by the sequences, and the lengths of the motifs were determined by the lowest *P*-value.

## Phylogenetic analysis

To infer a phylogenetic relationship, CNF00230 was selected as query protein sequence for BLASTp searches against the non-redundant database. Sequences resulting from these searches with cut-off criteria of >25% identity and e-value <10$^{-10}$ were considered homologous

proteins. The repeated clusters that consisted of protein sequences with the best hits to proteins from their own cluster were filtered out. The rest of the protein sequences (see Supplementary Data 5) were aligned using MUSCLE v3.8 with the default settings and then trimmed by trimAl v1.4 using the automated1 command. The maximum-likelihood method was used for the phylogenetic constructions. Maximum-likelihood analyses were performed by running 1000 bootstraps using RAxML-ng v0.9. The best-fitting amino acid substitution model was determined by IQ-TREE v2.1.2 and was chosen by the Bayesian information criterion. FigTree v1.4.4 was used to view the resulting trees, and the final layout was performed with Adobe Illustrator 2021.

### Protein structure predictions
The sequence of *C. deneoformans* Fmp1 (CNF00230) was obtained from FungiDB (https://fungidb.org/fungidb/app). Protein structure predictions were performed with AlphaFold2 version 2.1.1. A user-friendly interface for accessing AlphaFold2 has been made available by the Beijing Super Cloud Computing Center. For further analysis, structures with the best prediction quality were selected. Preparation of molecular graphic images was carried out in PyMOL version 2.5.0.

### Construction of functional TF networks and regulatory networks
The bipartite network that links TFs to the eight phenotypic traits was based on the results of phenotypic tests. Functional TF networks were visualized using Gephi software v0.9.3. Regulatory networks were visualized using Cytoscape software v3.8.0.

### Gene expression clustering
For temporal expression signature analyses, genes were filtered before clustering, and genes that did not show induced expression ($log_2TPM_{max}/TPM_{min} < 1$) or that exhibited persistently low expression levels (all TPM < 10) were screened out. The expression level was converted to a Z-score, and then hierarchical clustering was performed based on the temporal expression data, and 15 groups were obtained.

### Gene age estimation
Orthogroups were used to assign relative gene ages according to a previous study[72]. Briefly, protein sequences of 109 whole genomes across Basidiomycota and Ascomycota were downloaded from the JGI genome portal. MMseqs (2.0) was used to identify orthogroups with a sensitivity setting of 5.7. The coverage cut-off value was 0.2 and the e-value cutoff was set to 0.001. Relative gene ages were measured based on phylostratigraphic definitions.

### Protein expression and purification
Constructs directing the expression of recombinant proteins were generated by PCR using cDNA from XL280α as a template. Amplified DNA fragments coding for truncated Fmp1 (735–1690 aa) were inserted into vector pKLD116 with restriction enzymes Nco1 and Not1 to obtain plasmids directing the expression of MBP- and His-tagged recombinant proteins. An amplified DNA fragment coding for full-length Cpk1 was inserted into vector pET28a with restriction enzymes Nde1 and BamH1 to obtain a plasmid directing the expression of His-tagged protein. Plasmids were transformed into Rosetta (DE3) *E. coli*, and proteins were expressed at 18 °C. All proteins were expressed by MerryBio Co.,Ltd. Proteins were purified using Ni-NTA and dialyzed into 1 × PBS, 5% glycerol, pH 7.4.

### Pull-down assay
MBP-Fmp1 was incubated with Cpk1 in 200 μL buffer A (20 mM HEPES pH 7.5, 100 mM NaCl, 7.5 mM MgCl$_2$, 2 mM 2-mercaptoethanol, 50 μg/ mL BSA and 0.1% NP40) at room temperature for 1 h. Then, 20 μL of MBP-agarose beads (mbta-20, Chromotek, 1:200 dilution) were added to the reaction mixtures, followed by a 1 h incubation at room temperature under rotation. Following two washes, the proteins were eluted from the beads with 1× SDS-PAGE loading buffer. A pull-down experiment with unfused MBP was used as a control for non-specific binding to MBP.

For Western blot analyses, protein samples were separated by 10% SDS-PAGE and transferred electrophoretically to nitrocellulose membranes. Protein were detected with mouse anti-FLAG monoclonal antibody (F1804, Sigma-Aldrich, 1:1000 dilution) or MBP tag monoclonal antibody (66003-1-Ig, Proteintech, 1:1000 dilution). An HRP-conjugated anti-mouse antibody (BE0141, EasyBio, 1:10,000 dilution) was used as the secondary antibody. Antibody binding was visualized using SuperSignal West Pico Chemiluminescent Substrate (Pierce), according to the manufacturer's protocol.

### Analysis of susceptibility to chemicals and antifungal agents
Analyses of susceptibility to the presence of antifungal drugs and other stressors were performed as previously described[38]. Each strain was incubated at 30 °C in YPD medium for 12 h and diluted to $A_{600} = 1.0$ in distilled water (ddH$_2$O), whereupon five-fold dilutions were created for subsequent use. To analyze growth phenotypes at distinct temperatures, cells were spotted on solid YPD medium and monitored at a range of temperatures (4 °C, 25 °C, 30 °C, 37 °C, and 39 °C). To analyze susceptibility to stress, the prepared cells were spotted on YP or YPD medium containing one of the following chemicals: sorbitol to induce osmotic stress; NaCl or KCl to induce salt stress; hydrogen peroxide, tert-butyl hydroperoxide or menadione to induce oxidative stress; CoCl$_2$ to induce hypoxic stress; SDS, calcofluor white or Congo red to induce cell membrane/cell wall stress; tunicamycin or dithiothreitol to induce endoplasmic reticulum stress; UV to induce genotoxic stress; or antifungal agents (amphotericin B, fluconazole, 5-flucytosine, itraconazole or rapamycin).

### Statistical analysis
Statistical analyses were performed using R statistical platform, version 3.4.2. We used a two-tailed unpaired Student's *t* test to compare the mean florescence intensity, basidial maturation score or transcript levels between two groups. Fisher's exact tests were utilized to evaluate the significance between two sets of genes. A two-sided $P < 0.05$ was considered significant, and $P < 0.001$ was considered very significant. The data are shown as means ± SD from three or more independent experiments.

### Ethics statement
All experiments involving mice were performed under the guidance of "the regulation of the Institute of Microbiology, Chinese Academy of Sciences of Research Ethics Committee." The mouse models and procedures performed have been approved by the Institute of Microbiology, Chinese Academy of Sciences of Research Ethics Committee (Permit No. APIMCAS2021146).

### Reporting summary
Further information on research design is available in the Nature Portfolio Reporting Summary linked to this article.

## Data availability
All data needed to evaluate the conclusions in the paper are present in the paper or in Supplementary Information. The sequencing data has been deposited at the Gene Expression Omnibus (GEO) with accession number GSE135566. Source data are provided with this paper.

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

## Acknowledgements

We would like to thank Dr. Alexander Idnurm, Dr. Ci Fu, Dr. Cheng Gao, Dr. Chaoyang Xue, Dr. Qi Wu, Dr. Sheng Sun, and Dr. Youbao Zhao for critical reading and helpful suggestions. We thank Dr. Ci Fu from the lab of Dr. Joseph Heitman for important advice about *C. neoformans* blastospore dissection. We thank Dr. Tong Zhao and Dr. Chunli Li for their technical assistance in FACS and SEM analyses, respectively. This study was financially supported by the National Key Research and Development Program of China (2021YFC2300400 [L.W.], 2021YFA0911300 [X.L.], 2022YFC2303000 [L.W.]); the National Natural Science Foundation of China (Grants 32100153 [P.H.], 32270206 [X.T.]); and the CAS Interdisciplinary Innovation Team (L.W.). The funders had no role in study design, data collection or analysis, decision to publish, or preparation of the manuscript.

## Author contributions

All authors contributed to the data analysis. P.H., H.L., L.C., G.-J.H., X.T., X.Y., C.T., E.Y., and L.W. designed the experiments. P.H, G.-J.H., H.L., P.Z., X.Y., and Y.P., constructed most of the deletion mutants and conducted the phenotypic assays. H.D., L.C., C.T., Z.S., and E.Y., conducted most of the bioinformatics assays. Y.Y., performed the in vitro pull-down assays. X.T. conducted most of the RNA extraction and qRT-PCR. H.L. performed the scanning electron microscopy experiments. P.H., H.L., and P.Z., conducted most of FACS analysis. W.K. performed the animal experiments. L.W., X.L., E.Y., and G.L., contributed reagents/materials/analysis tools. L.W., P.H., H.L., H.D., and X.T., wrote the paper with contributions from all other authors.

## Competing interests

The authors declare no competing interests.

## Additional information

**Supplementary information** The online version contains
supplementary material available at

Linqi Wang.

**Peer review information** *Nature Communications* thanks the anon-
ymous reviewers for their contribution to the peer review of this
work. Peer reviewer reports are available.

