## [Peer Review File · Nature Communications]

Regulatory basis for reproductive flexibility in a meningitis-causing fungal pathogenReviewer #1 (Remarks to the Author):

This is a broad genetic study to analyze the basis for an interesting type of selfing in a fungal pathogen that infects immunocompromised patients. The general approach is to delete a large panel of genes encoding candidate transcription factors, phenotype these, and to focus on those that have an impact on sexual reproduction. A similar approach analyzing transcription factor mutants globally was published in Nature Communications in 2015 by Jung et al, cited here as ref 38. This study builds on those earlier results to focus primarily on sexual reproduction. The studies are also done in different isolate from a species that is different from the Jung et al study. Thus the two can be viewed as complementary. The studies presented here are conducted in the lab isolated XL280 strain that undergoes robust self-reproduction and which has served as a model for this. This strain was recently reported to have a lab acquired mutation in the ric8 G protein regulator (Sun et al PNAS 2022). This study was not cited here, or considered, and it would make sense to at least present and discuss this recent finding and how this may influence interpretation of these findings.

The paper would benefit from editing throughout for language and grammar. It is overall well written and clear, but would benefit from polishing the text via editing.

111 transcription factors were studied and there were 252 verified mutants. Were phenotypes congruent for independent isolates? In all cases? Were there cases where more than 2 mutants were isolated, and is that why there are more than 222 mutants? 7 transcription factors emerged as important, but one of them Mat2 is already known to be the key pheromone activated transcription factor. It is indicated in Figure 3A as not detected in the fusion assay? Is this due to a complete block in cell-cell fusion? If so, this could be clarified. What is the role or relationship of the other six factors to Mat2? This section seemed to lack a firm conclusion or outcome from this large amount of effort.

The authors describe at several points sex-determination genes. It was not clear if they are referring to pheromones, receptors, homeodomain proteins, all of these, or something else. It would help to clarify this.

By some type of expression clustering approach, the authors determined that a gene of unknown function called Fmp1 might be relevant for mating. They mutated it in *Cryptococcus* and also a related protein in the very distantly related fungus *N. crassa* and found the mutation impacts mating. By domain analysis they found a few different domains are harbored by this protein, and made the intuitive leap that this might represent a scaffold protein, similar conceptually to Ste5 in *S. cerevisiae*. They show MBP fusion protein data supporting that a MAP kinase called Cpk1 may physically interact with Fmp1. Thus, this is a candidate scaffold, but the evidence as presented falls short of being conclusive. Scaffolds typically function to bring sequential components of a pathway together, so in this case that might involve Ste7 and Ste11 as well as Cpk1.

The next section of the manuscript explores the possible roles of a quorum sensing system the authors have studied previously in the context of unisexual reproduction. They studied both a transcription factor that activates the system and the quorum sensing peptide and found that these may contribute to promote auto-diploidization, along with 49 other transcription factors that also impacted auto-diploidization. This seems like an extremely large number of factors to be regulating such a specific process, and it seems likely that the phenotypes for many of these mutants may be quite indirect. To further link the Cqs2 factor to sexual development, ChIP-seq studies were conducted to define direct targets, and this identified Pum1, a pumilio family RNA binding protein. Deletion of the PUM1 gene led to defects in development and over-expression promoted formation of diploid cells, linking it to the pathway. The authors then focused on 13 candidate targets that based on RNA-seq were upregulated by Cqs2 and by Pum1.

There is a long section seeking to address why Pum1 might promote haploid to diploid but not diploid to higher ploidy transitions, and this seemed to be a distraction from this line of investigation. It is not clear how much these are adding to the clarity of presentation, and streamlining this would help marshal the more important findings and be less distracting to the reader.

The authors were led to focus on 7 genes regulated by Pum1 in the haploid context, succeeded in deleting five of them, and one of these abolished production of diploid blastospores and also blocked the ability of Pum1 overexpression to drive ploidy increase. This gene was named CUA1. Interestingly, it encodes a very large protein >350 kDa, and a tagged version of this protein co-localized with a spindle pole body marker. This leads the authors to propose that this protein may be promoting haploid to diploid transition by impacting SPB function, which is an interesting hypothesis.

In summary, the authors have marshaled deletion analysis, RNA-seq, ChIP, and protein localization and interaction studies to identify a series of candidate factors that contribute to self sexual development. This represents a tremendous amount of work, and is several studies rolled into one. Several of these require further analysis to validate and understand the mechanisms involved, such as for the candidate scaffold Fmp1 and the possible SPB component or regulator Cua1.

Reviewer #2 (Remarks to the Author):

The manuscript explores the transcription factors involving in the two mating systems found in the human pathogenic fungus *Cryptococcus neoformans*. The research has a number of components, roughly divided into (i) deletion of 111 predicted transcription factors that are regulated during mating and characterizing their phenotypes; (ii) identification of ancient or 'hub' components involved in mating, including a new component that is conserved across fungi and show to also function in mating in *Neurospora crassa* [a noteworthy strength was in exploring similar gene functions in a distant relative], and (iii) defining how the autopolyploidization process works for unisexual mating via the discovery of a new component CUA1.

This is a huge body of work and features a wonderful set of illustrations of the data.

Perhaps the one criticism in trying to report so much that the overall message from the work is harder to see. In some ways the mutant collection is similar to what have been described as 'resource collections' for *C. neoformans* (e.g. Hiten Madhani and Yong-Sun Bahn labs), and then two distinct stories emerge as examples of what can be identified in more detailed analysis, one on bisexual mating and the discovery of FMP1, and the second on unisexual mating and discovery of CUA1. Hence, the authors might like to consider whether a different style of explaining the overall research direction and findings would make it clearer to readers, especially those outside of the field of *Cryptococcus* mating systems.

Points for discussion/consideration

- It was curious that none of the 111 transcription factors were essential for viability, or are these those selected that could be disrupted?
- An alternative approach to mutating transcription factor genes upregulated during mating could have been to constitutive express those turned down, to address concepts around the commitment to asexual vs. sexual reproduction, or potential suppressors of autopolyploidization.
- One weakness is that the screen of the 111 transcription factor mutants is likely only capturing genes involved in early steps of mating, i.e. up to cell fusion. It is likely that if a set of mutants in the opposite mating type were available to perform mutant x mutant crosses, additional genes would have emerged. That is probably beyond the scope of this work.
- How much overlap is there with this collection and that created by Bahn et al., in terms of the transcription factor genes included and in vitro phenotypes?
- For the phylogeny in figure 4a (and species in table S5), it may be useful to elaborate on the level of conservation. That is, are homologs of FMP1 found beyond these species, or is it absent from some fungi (and if so, what fulfills its place)?

Minor typographical points or rewording are as follows.

Line 70: 'differentiate to form a sexual'.

Line 124: 'In addition, a new'.

Line 261: can likely delete 'importantly'.

Line 619: '40 mM'.

Lines 629-634: what was the origin of the wild type and mutant strains of *N. crassa*? There did not seem to be any details here. Perhaps from the *N. crassa* deletion collection.

Reference section: it may be necessary to add italics to species and gene names. A number of the electronic articles are missing article numbers (equivalent of page numbers).

Line 1076: update citation.

REVIEWER COMMENTS

Reviewer #1 (Remarks to the Author):

1. This is a broad genetic study to analyze the basis for an interesting type of selfing in a fungal pathogen that infects immunocompromised patients. The general approach is to delete a large panel of genes encoding candidate transcription factors, phenotype these, and to focus on those that have an impact on sexual reproduction. A similar approach analyzing transcription factor mutants globally was published in Nature Communications in 2015 by Jung et al, cited here as ref 38. This study builds on those earlier results to focus primarily on sexual reproduction. The studies are also done in different isolate from a species that is different from the Jung et al study. Thus the two can be viewed as complementary. The studies presented here are conducted in the lab isolated XL280 strain that undergoes robust self-reproduction and which has served as a model for this. This strain was recently reported to have a lab acquired mutation in the *ric8 G* protein regulator (Sun et al PNAS 2022). This study was not cited here, or considered, and it would make sense to at least present and discuss this recent finding and how this may influence interpretation of these findings.

Response:

We thank the reviewer for the helpful comments. XL280 α is one of the most widely employed isolates in studies of *Cryptococcus* sexual reproduction, especially unisexual reproduction (selfing). The robust unisexual reproduction observed in this strain has been shown to be attributable to the *RIC8*^{G1958T} loss-of-function mutation¹, as reported by the study mentioned by this reviewer; due to the importance of this study, it has been cited and discussed in the updated version of the manuscript.

It would be very difficult to assess the epistatic genetic interactions of this mutation with all selfing genes identified in this study. Nonetheless, we performed new experiments to evaluate the influence of this mutation on the function of *Cqs2* and *Pum1* (new Figs. S4e, f and S5e, f), which, in XL280 α , are the critical factors coordinating the multistage unisexual life cycle or inducing unisexual autodiploidization, respectively (Fig. 5, 6). Unexpectedly, overexpression of *CQS2* in another *C. deneoformans* strain, JEC21 α , which does not harbor the *RIC8*^{G1958T} allele¹, enhanced the robustness of self-filamentation (new Fig. S4e, f). Furthermore, the JEC21 α -derived *CQS2* overexpression strain produced robust sporulated basidia after 2 weeks of incubation on V8 agar, while sporulation was undetected in its parent strain (new Fig. S4e, f). The considerable promotion of unisexual sporulation, the last phase of the unisexual cycle, in the overexpression strain suggested that *Cqs2* is important in the induction and orchestration of the sequential selfing stages in the JEC21 α background. Likewise, enhanced expression of *Pum1* in JEC21 α resulted in a remarkable increase in cell size and a prominent population of diploid cells based on a FACS-based ploidy evaluation (new Fig. S5e, f). All of these results recapitulated our findings in XL280 α (Figs. 5d-e and 6c-d). Taken together, these results suggest that the key roles of *Cqs2* and *Pum1* in unisexual reproduction are not unique to XL280 α but that they are conserved in different *C. deneoformans* isolates. These new data and the corresponding descriptions have been incorporated into the updated version of the

manuscript.

2. The paper would benefit from editing throughout for language and grammar. It is overall well written and clear, but would benefit from polishing the text via editing.

Response:

We thank the reviewer for the suggestion. The manuscript has been thoroughly edited for language and grammar by a native English speaker.

3. 111 transcription factors were studied and there were 252 verified mutants. Were phenotypes congruent for independent isolates? In all cases? Were there cases where more than 2 mutants were isolated and is that why there are more than 222 mutants? 7 transcription factors emerged as important, but one of them Mat2 is already known to be the key pheromone activated transcription factor. It is indicated in Figure 3A as not detected in the fusion assay? Is this due to a complete block in cell-cell fusion? If so, this could be clarified. What is the role or relationship of the other six factors to Mat2? This section seemed to lack a firm conclusion or outcome from this large amount of effort.

Response:

To accurately validate the phenotype and exclude unlinked mutational effects, we generated at least two independent transcription factor (TF) mutants for all 111 TFs. When two TF mutants showed inconsistent phenotypes, additional TF mutants were generated to exclude outlier mutants. Thus, of the 111 TFs, 81 TFs had two independent mutants, and 30 TFs had three independent mutants, bringing a total of 252 mutants. This information has been added to the Materials and Methods section in the updated version of the manuscript.

This reviewer is correct. Our mating efficiency analysis did show that deletion of *MAT2* caused a complete defect in bisexual cell-cell fusion in *C. deneoformans*, resulting in a failure of this fungus to produce fusants, as indicated by “not detected” in Figure 3A. This result is consistent with previous studies on *C. deneoformans* that demonstrate that Mat2 is a master regulator of bisexual cell-cell fusion²⁻⁴. In the revised manuscript, we have added a sentence in the legend of Figure 3A to avoid any possible confusion.

Our functional genomics strategy identified seven transcription factors that are important for bisexual mating, including Mat2. According to our transcriptomic data, among these transcription factors, only the deletion of Hob7 resulted in a significant change in the mRNA levels of *MAT2* ($\log_2(\text{fold change}) = 1.2$, $P_{adj} = 1 \times 10^{-19}$), but no significant effects of other transcription factors on *MAT2* expression were detected (Supplementary Table 3), implying that most of the syngamy TFs may collectively mediate a coordinated regulation of a large number of mating-responsive genes. In agreement with this idea, the gene accumulation curve showed a remarkable difference in the mating-responsive genes regulated by these TFs, supporting that they act in concert in mediating transcriptional response to bisexual mating in *C. deneoformans* (Fig. 3c). In the updated version of the manuscript, we have added additional descriptions to emphasize this conclusion.

4. The authors describe at several points sex-determination genes. It was not clear if they are referring to pheromones, receptors, homeodomain proteins, all of these, or something else. It would help to clarify this.

Response:

According to previous studies, the sex-determining genes in *C. deneoformans* mating type (MAT) α strains are genes encoding pheromones (*MF α*), pheromone receptor (*STE3 α*), and a homeodomain regulator (*SXII α*)⁵⁻⁸. In the updated version of the manuscript, we have clarified the sex-determining genes in the main text and cited the related references.

5. By some type of expression clustering approach, the authors determined that a gene of unknown function called Fmp1 might be relevant for mating. They mutated it in *Cryptococcus* and also a related protein in the very distantly related fungus *N. crassa* and found the mutation impacts mating. By domain analysis they found a few different domains are harbored by this protein, and made the intuitive leap that this might represent a scaffold protein, similar conceptually to Ste5 in *S. cerevisiae*. They show MBP fusion protein data supporting that a MAP kinase called Cpk1 may physically interact with Fmp1. Thus, this is a candidate scaffold, but the evidence as presented falls short of being conclusive. Scaffolds typically function to bring sequential components of a pathway together, so in this case that might involve Ste7 and Ste11 as well as Cpk1.

Response:

We agree with this reviewer that scaffold proteins, such as Ste5, function to bring sequential components of a pathway together. In *Saccharomyces cerevisiae*, Ste5 binds the MAPKKK (Ste11), the MAPKK (Ste7), and the MAPK (Fus3, the homologue of *C. deneoformans* Cpk1) of the mating MAPK cascade⁹⁻¹¹. Following the suggestion of this reviewer, we expressed *C. deneoformans*-derived Ste7 in *Escherichia coli* (*E. coli*) and performed new *in vitro* pull-down experiments with the purified proteins. We did not detect a physical interaction between Ste7 and Fmp1 (author response image 1), which was shown to bind Cpk1 (Fig. 4j). These data indicate a difference in molecular function between Ste5 and Fmp1, although they are required for mating and share functional domains⁹⁻¹¹(Figs. 4b, e and f). Thus, Ste5 and Fmp1 are not analogs of each other, consistent with the remarkable difference in their sequences and structures (Fig. S3). As there is no compelling evidence to support Fmp1 as a scaffold protein, we have revised the corresponding statements accordingly to avoid misleading readers.

Author response image 1. Fmp1 might not interact with Ste7. a. Ste7-HA was expressed in Rosetta

E. coli and purified via affinity chromatography using Ni-NTA resin. The protein was analyzed by SDS-PAGE followed by Coomassie staining. BSA, bovine serum albumin. **b.** MBP pull-down assays were performed to investigate an interaction between MBP-Fmp¹⁷³⁵⁻¹⁶⁹⁰ and Ste7-HA. Purified MBP-Fmp¹⁷³⁵⁻¹⁶⁹⁰ or MBP (control) proteins were used to pull down Ste7-HA using MBP-agarose beads. Anti-HA and anti-MBP antibodies were used for the immunoblotting.

6. The next section of the manuscript explores the possible roles of a quorum sensing system the authors have studied previously in the context of unisexual reproduction. They studied both a transcription factor that activates the system and the quorum sensing peptide and found that these may contribute to promote auto-diploidization, along with 49 other transcription factors that also impacted auto-diploidization. This seems like an extremely large number of factors to be regulating such a specific process, and it seems likely that the phenotypes for many of these mutants may be quite indirect. To further link the Cqs2 factor to sexual development, ChIP-seq studies were conducted to define direct targets, and this identified Pum1, a pumilio family RNA binding protein. Deletion of the PUM1 gene led to defects in development and over-expression promoted formation of diploid cells, linking it to the pathway. The authors then focused on 13 candidate targets that based on RNA-seq were upregulated by Cqs2 and by Pum1.

There is a long section seeking to address why Pum1 might promote haploid to diploid but not diploid to higher ploidy transitions, and this seemed to be a distraction from this line of investigation. It is not clear how much these are adding to the clarity of presentation, and streamlining this would help marshal the more important findings and be less distracting to the reader.

Response:

We thank the reviewer for the helpful suggestion. In the revised manuscript, we have streamlined the Abstract and Results sections related to the haploid-dependent ploidy duplication mechanism that is mediated by Pum1.

7. The authors were led to focus on 7 genes regulated by Pum1 in the haploid context, succeeded in deleting five of them, and one of these abolished production of diploid blastospores and also blocked the ability of Pum1 overexpression to drive ploidy increase. This gene was named CUA1. Interestingly, it encodes a very large protein >350 kDa, and a tagged version of this protein co-localized with a spindle pole body marker. This leads the authors to propose that this protein may be promoting haploid to diploid transition by impacting SPB function, which is an interesting hypothesis.

In summary, the authors have marshaled deletion analysis, RNA-seq, ChIP, and protein localization and interaction studies to identify a series of candidate factors that contribute to self sexual development. This represents a tremendous amount of work, and is several studies rolled into one. Several of these require further analysis to validate and understand the mechanisms involved, such as for the candidate scaffold Fmp1 and the possible SPB component or regulator Cua1.

Response:

We thank the reviewer for the summary of our work and positive comments. In this study, we revealed the regulatory basis that shapes the different sexual life cycles in *C. deneoformans*, a model organism for the study of *Cryptococcus* sexual reproduction. As

mentioned by this reviewer, we identified a series of novel factors contributing to the *Cryptococcus* sexual cycles, including Fmp1 and Cua1. We showed that Fmp1 directly interacts with Cpk1, the terminal kinase of the *Cryptococcus* mating MAPK cascade, and we showed that its coding gene displayed a highly inducible expression (approximately 48-fold) in response to mating cues. To further understand the function of Fmp1, we performed new experiments to examine its ability to bind Ste7, the upstream kinase of Cpk1. In these experiments, we did not detect a physical interaction between Ste7 and Fmp1, as illustrated by author response image 1. Thus, Fmp1 appears not to share functional similarity with Ste5, the well-known mating scaffold protein, despite their sharing of identical domains and domain arrangement⁹⁻¹¹(Fig. 4e). These data support the idea that Fmp1 and its orthologues represent previously unidentified components of the mating MAPK pathway in *C. deneoformans* or other fungal species, as they are important in opposite-sex mating in several different fungi, as revealed by our genetic and phenotypic analyses (Fig. 4b-d).

Cua1 was identified to play an important role in the unisex-specific event, pre-meiotic autodiploidization, but it is largely dispensable for bisexual syngamy or *Cryptococcus* growth under 27 *in vitro* and *in vivo* conditions (Fig. 7b-e). These data highlight its functional specificity in unisexual autodiploidization. We further showed that Cua1 acts as a spindle pole body (SPB)-associated protein, according to its co-localization with an indicator protein of *Cryptococcus* SPB (Fig. 7g, h). Of note, disturbing SPB assembly serves as an important mechanism for ploidy increase in different eukaryotic species¹²⁻¹⁵. For example, in *S. cerevisiae*, a dysfunctional SPB has been shown to be able to nucleate microtubules from the cytoplasmic side but not from the nuclear side, resulting in a nondisjunction event and spontaneous diploidization⁹. These previous findings suggest a possible connection between SPB function and unisexual autodiploidization. Revealing the mechanisms by which the SPB-associated protein Cua1 activates autodiploidization during selfing in *C. deneoformans* or other *Cryptococcus* species will be an interesting direction for future investigation.

Reviewer #2 (Remarks to the Author):

The manuscript explores the transcription factors involving in the two mating systems found in the human pathogenic fungus *Cryptococcus deneoformans*. The research has a number of components, roughly divided into (i) deletion of 111 predicted transcription factors that are regulated during mating and characterizing their phenotypes; (ii) identification of ancient or 'hub' components involved in mating, including a new component that is conserved across fungi and show to also function in mating in *Neurospora crassa* [a noteworthy strength was in exploring similar gene functions in a distant relative], and (iii) defining how the autoployploidization process works for unisexual mating via the discovery of a new component CUA1.

This is a huge body of work and features a wonderful set of illustrations of the data.

Perhaps the one criticism in trying to report so much that the overall message from the work is harder to see. In some ways the mutant collection is similar to what have been described as

‘resource collections’ for *C. neoformans* (e.g. Hiten Madhani and Yong-Sun Bahn labs), and then two distinct stories emerge as examples of what can be identified in more detailed analysis, one on bisexual mating and the discovery of FMP1, and the second on unisexual mating and discovery of CUA1. Hence, the authors might like to consider whether a different style of explaining the overall research direction and findings would make it clearer to readers, especially those outside of the field of *Cryptococcus* mating systems.

Response:

We thank the reviewer for thoroughly examining our manuscript and providing important comments to guide our revision. We have streamlined the Abstract and Results sections related to the haploid-dependent ploidy duplication mechanism mediated by *Pum1* to make the logic clearer. Responses to other comments are provided below.

Points for discussion/consideration

1. It was curious that none of the 111 transcription factors were essential for viability, or are these those selected that could be disrupted?

Response:

Re-analysis of two publicly available sets of time-series transcriptomic data revealed that in addition to the 111 transcription factor (TF) genes, four TF genes, including *HSF1*, *CDC39*, *PZF1*, and *ESAI*, displayed dynamic expression during sexual development. Their homologs have been demonstrated to be essential genes in other fungal species¹⁶⁻²¹. Consistently, we failed to obtain their corresponding mutants after repeated attempts. Thus, they were not included in our study. We have added the corresponding information in the text of the updated version of the manuscript.

2. An alternative approach to mutating transcription factor genes upregulated during mating could have been to constitutive express those turned down, to address concepts around the commitment to asexual vs. sexual reproduction, or potential suppressors of autopolyploidization.

Response:

We thank the reviewer for this suggestion. As mentioned earlier, the 111 transcription factor genes were chosen in our study based on their dynamic expression in sexual development. These genes included some that are transcriptionally up-regulated and some that are down-regulated during sexual development. Our phenotypic analysis identified a number of ‘repressors’ whose absence resulted in enhanced phenotypes (Fig. 1d). Nevertheless, we agree with this reviewer about the importance of the forced expression of TF genes that are turned down during sexual life cycles. This approach can be treated as complementary to our current systematic method that mainly assessed the impact of the deletion of TF genes on sexual regulation, and it warrants further investigation.

3. One weakness is that the screen of the 111 transcription factor mutants is likely only capturing genes involved in early steps of mating, i.e. up to cell fusion. It is likely that if a set of mutants in the opposite mating type were available to perform mutant x mutant crosses, additional genes

would have emerged. That is probably beyond the scope of this work.

Response:

The α -a cell-cell fusion event is an early event of bisexual reproduction in *C. deneoformans*, and it is also bisex-specific, while mid- and late-sexual processes are highly similar in bisexual and unisexual life cycles. These processes include hyphal initiation and elongation, basidial maturation, and sporulation (Fig. 1a). Our phenotypic assays identified not only the TFs engaged in α -a cell-cell fusion (Figs. 1d and 3a), but also 186 genotype-phenotype associations related to mid- and late-sexual processes (Fig. 1d).

For the assessment of α -a cell-cell fusion, we mainly used unilateral mating analysis. With this approach, we reproduced the phenotypes of almost all TFs whose effects on α -a cell fusion were documented in *C. deneoformans* or other *Cryptococcus* pathogens (Fig. 1d)^{2,4,21-32}. This suggests that our approach is reliable in identifying TF genes that are critical for this sexual event. We also agree with the possibility raised by this reviewer that bilateral mating analysis might reveal more genes involved in sexual regulation, especially those that play fine-tuning roles.

4. How much overlap is there with this collection and that created by Bahn et al., in terms of the transcription factor genes included and in vitro phenotypes?

Response:

A total of 76 TFs were identified as homologs by comparing the TFs included in the *C. deneoformans* mutant library that we generated with those in the *C. neoformans* mutant collection created by Bahn *et al.*²¹. The *C. neoformans* mutant collection was derived from the H99 isolate, which has a much weaker ability to undergo sexual differentiation compared with *C. deneoformans* strain XL280a. In addition, to date, a unisexual life cycle has not been observed in the *C. neoformans* H99 strain. Thus, by using the *C. neoformans* TF mutant collection, a previous study revealed only the TFs involved in bisex (mostly bisexual filamentation), but not those in unisex, identifying 37 TFs and 44 genotype-phenotype associations related to bisexual filamentation or α -a cell-cell fusion²¹. Homologs of these 37 sexual TFs were included in our study, and importantly, our phenotypic assays recapitulated all 44 genotype-phenotype associations (Fig. 1d). Furthermore, we also identified 312 genotype-phenotype connections, which have yet to be reported in *C. deneoformans* or other *Cryptococcus* species, including *C. neoformans*.

5. For the phylogeny in figure 4a (and species in table S5), it may be useful to elaborate on the level of conservation. That is, are homologs of FMP1 found beyond these species, or is it absent from some fungi (and if so, what fulfills its place)?

Response:

Using BLAST analyses, we found that Fmp1 is widely distributed in species belonging to Basidiomycota and Ascomycota, except for some species from *Saccharomycotina* that harbor the genes encoding Ste5 and its orthologues. The homologs of Ste5 and Fmp1 share the PH, RING and vWA domains (Fig. 4e)⁹⁻¹¹. These domains have been documented to be responsible for protein-protein interactions⁹⁻¹¹. Accordingly, Ste5 and Fmp1 can bind the terminal kinases of the mating MAPK cascade in *S. cerevisiae* (Fus3) and *C.*

deneoformans (Cpk1) (Fig. 4j)⁹⁻¹¹, respectively. However, reciprocal BLAST searches did not demonstrate an orthologous relationship, suggesting that Fmp1 and Ste5 are evolutionarily remote proteins or are not homologous to each other. Supporting this idea, the structure of the vWA domain, which is essential for the scaffolding function of Ste5, is markedly different from that of the corresponding domain in Fmp1, as predicted by the AlphaFold 2 program (Fig. S3). Furthermore, to address question 5 raised by reviewer #1, we tested the ability of Fmp1 to bind another kinase of the mating MAPK pathway, Ste7, whose homologue in *S. cerevisiae* has been proven to interact with Ste5. However, when purified *C. deneoformans* Ste7 and Fmp1 were subjected to *in vitro* pull-down assays, no such interaction was observed (author response image 1). These data indicate that Ste5 and Fmp1 may exert different molecular functions, even though they have similarities in terms of their functional domains and their interactions with the terminal kinase of the mating MAPK cascade.

Minor typographical points or rewording are as follows.

6. Line 70: 'differentiate to form a sexual'.

Response:

Changed.

7. Line 124: 'In addition, a new'.

Response:

Changed.

8. Line 261: can likely delete 'importantly'.

Response:

Changed.

9. Line 619: '40 mM'.

Response:

Changed.

10. Lines 629-634: what was the origin of the wild-type and mutant strains of *N. crassa*? There did not seem to be any details here. Perhaps from the *N. crassa* deletion collection.

Response:

We apologize for the omission of this information, which has now been added in the Materials and Methods section in the updated version of the manuscript. It reads as follows: "The *N. crassa* wild-type and mutant strains were obtained from the *N. crassa* deletion collection at the Fungal Genetics Stock Center (FGSC)³³ and knockout event of the mutant was confirmed by PCR".

11. Reference section: it may be necessary to add italics to species and gene names. A number of the electronic articles are missing article numbers (equivalent of page numbers).

Response:

We thank the reviewer for pointing out these errors. In the revised manuscript, we have corrected all of these errors.

12. Line 1076: update citation.

**Response:
updated.**

Reference

1. Sun, S., Roth, C., Floyd Averette, A., Magwene, P.M. & Heitman, J. Epistatic genetic interactions govern morphogenesis during sexual reproduction and infection in a global human fungal pathogen. *Proc Natl Acad Sci U S A* **119**(8):e2122293119 (2022).
2. Feretzaki, M. & Heitman, J. Genetic circuits that govern bisexual and unisexual reproduction in *Cryptococcus neoformans*. *PLoS Genet* **9**, e1003688 (2013).
3. Kruzel, E.K., Giles, S.S. & Hull, C.M. Analysis of *Cryptococcus neoformans* sexual development reveals rewiring of the pheromone-response network by a change in transcription factor identity. *Genetics* **191**, 435-49 (2012).
4. Lin, X., Jackson, J.C., Feretzaki, M., Xue, C. & Heitman, J. Transcription factors Mat2 and Znf2 operate cellular circuits orchestrating opposite- and same-sex mating in *Cryptococcus neoformans*. *PLoS Genet* **6**, e1000953 (2010).
5. Stanton, B.C., Giles, S.S., Staudt, M.W., Kruzel, E.K. & Hull, C.M. Allelic exchange of pheromones and their receptors reprograms sexual identity in *Cryptococcus neoformans*. *PLoS Genet* **6**, e1000860 (2010).
6. Ni, M., Feretzaki, M., Sun, S., Wang, X. & Heitman, J. Sex in fungi. *Annu Rev Genet* **45**, 405-30 (2011).
7. Fu, C., Sun, S., Billmyre, R.B., Roach, K.C. & Heitman, J. Unisexual versus bisexual mating in *Cryptococcus neoformans*: Consequences and biological impacts. *Fungal Genet Biol* **78**, 65-75 (2015).
8. Yan, Z. *et al.* Deletion of the sex-determining gene *SXII*alpha enhances the spread of mitochondrial introns in *Cryptococcus neoformans*. *Mob DNA* **9**, 24 (2018).
9. Davis, R.J. Cell biology. A scaffold switch to insulate. *Science* **337**, 1178-9 (2012).
10. Bhattacharyya, R.P. *et al.* The Ste5 scaffold allosterically modulates signaling output of the yeast mating pathway. *Science* **311**, 822-6 (2006).
11. Good, M.C., Zalatan, J.G. & Lim, W.A. Scaffold proteins: hubs for controlling the flow of cellular information. *Science* **332**, 680-6 (2011).
12. Sing, T.L. *et al.* The budding yeast RSC complex maintains ploidy by promoting spindle pole body insertion. *J Cell Biol* **217**, 2445-2462 (2018).
13. Chial, H.J., Giddings, T.H., Jr., Siewert, E.A., Hoyt, M.A. & Winey, M. Altered dosage of the *Saccharomyces cerevisiae* spindle pole body duplication gene, *NDC1*, leads to aneuploidy and polyploidy. *Proc Natl Acad Sci U S A* **96**, 10200-5 (1999).
14. Toma, M.I. *et al.* Correlation of centrosomal aberrations with cell differentiation and DNA ploidy in prostate cancer. *Anal Quant Cytol Histol* **32**, 1-10 (2010).
15. Jiang, F. *et al.* Centrosomal abnormality is common in and a potential biomarker for bladder cancer. *Int J Cancer* **106**, 661-5 (2003).
16. Yang, D.H. *et al.* Rewiring of Signaling Networks Modulating Thermotolerance in the Human

- Pathogen *Cryptococcus neoformans*. *Genetics* **205**, 201-219 (2017).
17. Wiederrecht, G., Seto, D. & Parker, C.S. Isolation of the gene encoding the *S. cerevisiae* heat shock transcription factor. *Cell* **54**, 841-53 (1988).
 18. Collart, M.A. & Struhl, K. *CDC39*, an essential nuclear protein that negatively regulates transcription and differentially affects the constitutive and inducible *HIS3* promoters. *EMBO J* **12**, 177-86 (1993).
 19. Woychik, N.A. & Young, R.A. Genes encoding transcription factor *III A* and the RNA polymerase common subunit *RPB6* are divergently transcribed in *Saccharomyces cerevisiae*. *Proc Natl Acad Sci U S A* **89**, 3999-4003 (1992).
 20. Smith, E.R. *et al.* ESA1 is a histone acetyltransferase that is essential for growth in yeast. *Proc Natl Acad Sci U S A* **95**, 3561-5 (1998).
 21. Jung, K.W. *et al.* Systematic functional profiling of transcription factor networks in *Cryptococcus neoformans*. *Nat Commun* **6**, 6757 (2015).
 22. Liu, L. *et al.* Genetic basis for coordination of meiosis and sexual structure maturation in *Cryptococcus neoformans*. *Elife* **7**:e38683. (2018).
 23. Tian, X. *et al.* *Cryptococcus neoformans* sexual reproduction is controlled by a quorum sensing peptide. *Nat Microbiol* **3**, 698-707 (2018).
 24. Wang, L., Zhai, B. & Lin, X. The link between morphotype transition and virulence in *Cryptococcus neoformans*. *PLoS Pathog* **8**, e1002765 (2012).
 25. Fu, C., Donadio, N., Cardenas, M.E. & Heitman, J. Dissecting the Roles of the Calcineurin Pathway in Unisexual Reproduction, Stress Responses, and Virulence in *Cryptococcus deneoformans*. *Genetics* **208**, 639-653 (2018).
 26. Liu, H. *et al.* A Velvet Transcription Factor Specifically Activates Mating through a Novel Mating-Responsive Protein in the Human Fungal Pathogen *Cryptococcus deneoformans*. *Microbiol Spectr*, **10**(3):e0265321 (2022).
 27. Yue, C. *et al.* The *STE12*alpha homolog is required for haploid filamentation but largely dispensable for mating and virulence in *Cryptococcus neoformans*. *Genetics* **153**, 1601-15 (1999).
 28. Ren, P. *et al.* Transcription factor *STE12*alpha has distinct roles in morphogenesis, virulence, and ecological fitness of the primary pathogenic yeast *Cryptococcus gattii*. *Eukaryot Cell* **5**, 1065-80 (2006).
 29. Idnurm, A. & Heitman, J.J.P.B. Light Controls Growth and Development via a Conserved Pathway in the Fungal Kingdom. *PLoS Biol* **3**(4):e95 (2005).
 30. Jung, W.H. & Kronstad, J.W. The iron-responsive, GATA-type transcription factor Cir1 influences mating in *Cryptococcus neoformans*. *Mol Cells* **31**, 73-7 (2011).
 31. Davidson, R.C., Nichols, C.B., Cox, G.M., Perfect, J.R. & Heitman, J. A MAP kinase cascade composed of cell type specific and non-specific elements controls mating and differentiation of the fungal pathogen *Cryptococcus neoformans*. *Mol Microbiol* **49**, 469-85 (2003).
 32. Cramer, K.L., Gerrald, Q.D., Nichols, C.B., Price, M.S. & Alspaugh, J.A. Transcription factor Nrg1 mediates capsule formation, stress response, and pathogenesis in *Cryptococcus neoformans*. *Eukaryot Cell* **5**, 1147-56 (2006).
 33. McCluskey, K., Wiest, A. & Plamann, M. The Fungal Genetics Stock Center: a repository for 50 years of fungal genetics research. *J Biosci* **35**, 119-26 (2010).

Reviewer #1 (Remarks to the Author):

The reviewer appreciates the revisions and inclusion of additional data and the authors have addressed the questions that were raised. The issue of the main strain studied being a lab evolved variant with a mutation that activates the pathway remains, and suggests further study in natural isolates will be important along the lines of the additional data included here for one such isolate. The further analysis of the proposed scaffold showed that it doesn't interact with the second protein tested, so whether or not it serves as a scaffold (which by their very nature need to bind to at least two different proteins to bring them together) remains to be studied. Fine for this to be the next study rather than part of this one. This study advances the field and will be of interest to fungal geneticists broadly.

Reviewer #2 (Remarks to the Author):

The authors have addressed the points made by this reviewer.

Reviewer #1 (Remarks to the Author):

The reviewer appreciates the revisions and inclusion of additional data and the authors have addressed the questions that were raised. The issue of the main strain studied being a lab evolved variant with a mutation that activates the pathway remains, and suggests further study in natural isolates will be important along the lines of the additional data included here for one such isolate. The further analysis of the proposed scaffold showed that it doesn't interact with the second protein tested, so whether or not it serves as a scaffold (which by their very nature need to bind to at least two different proteins to bring them together) remains to be studied. Fine for this to be the next study rather than part of this one. This study advances the field and will be of interest to fungal geneticists broadly.

Response:

We thank the reviewer for the positive comments. We agree with the reviewer that further studies in natural isolates of *C. deneoformans* and determining the potential role of Fmp1 as a scaffold protein will be important.

Reviewer #2 (Remarks to the Author):

The authors have addressed the points made by this reviewer.

Response:

We thank the reviewer for the helpful suggestions and comments.